# Spread in climate policy scenarios unravelled

Mark M. Dekker[1,2✉], Andries F. Hof[1,2,3], Maarten van den Berg[1], Vassilis Daioglou[1,2], Rik van Heerden[1], Kaj-Ivar van der Wijst[2] & Detlef P. van Vuuren[1,2]

Analysis of climate policy scenarios has become an important tool for identifying mitigation strategies, as shown in the latest Intergovernmental Panel on Climate Change Working Group III report[1]. The key outcomes of these scenarios differ substantially not only because of model and climate target differences but also because of different assumptions on behavioural, technological and socio-economic developments[2–4]. A comprehensive attribution of the spread in climate policy scenarios helps policymakers, stakeholders and scientists to cope with large uncertainties in this field. Here we attribute this spread to the underlying drivers using Sobol decomposition[5], yielding the importance of each driver for scenario outcomes. As expected, the climate target explains most of the spread in greenhouse gas emissions, total and sectoral fossil fuel use, total renewable energy and total carbon capture and storage in electricity generation. Unexpectedly, model differences drive variation of most other scenario outcomes, for example, in individual renewable and carbon capture and storage technologies, and energy in demand sectors, reflecting intrinsic uncertainties about long-term developments and the range of possible mitigation strategies. Only a few scenario outcomes, such as hydrogen use, are driven by other scenario assumptions, reflecting the need for more scenario differentiation. This attribution analysis distinguishes areas of consensus as well as strong model dependency, providing a crucial step in correctly interpreting scenario results for robust decision-making.

Model projections play an important part in the recent reports of the Intergovernmental Panel on Climate Change (IPCC). Based on several assumptions on population and economic developments, these projections explore how different climate policies affect energy supply and demand, and how climate changes as a result. These scenarios are made available through large databases: the Coupled Model Intercomparison Project Phase 6 (CMIP6) database[6] with climate physics projections and the Sixth Assessment Report (AR6) database[7] with climate change mitigation scenarios, each of which shows a wide range of scenario outcomes. A core question for policymakers, researchers and other users of these projections is which elements of mitigation strategies are robust: that is, aspects such as technology roll-out, energy carriers and emission levels that are distinct for different climate goals, surpassing any other notable uncertainties. In physical climate science, the robust attribution of phenomena such as increased precipitation to different levels of global warming is a main topic, and although the topic is also important for mitigation literature[8]—for instance, to inform stakeholders on how the electricity mix differs between a 1.5 °C world and a 3 °C world—a comprehensive overview of robustness in mitigation strategies is, to our knowledge, still pending.

Three main drivers of the spread in climate policy scenarios can be distinguished: climate targets, model characteristics and scenario assumptions. Climate targets (or more precisely climate outcomes) are

an obvious driver: an energy system that achieves specific climate goals (such as those of the Paris Agreement) differs notably from a system that does not. The AR6 of the Working Group III (WGIII) of the IPCC[1] indicates that many key energy variables correlate with climate outcomes. The approximately 1,200 scenarios in the AR6 database are labelled with categories ranging from C1 (below 1.5 °C temperature change in 2100 with limited or no temperature overshoot) to C8 (above 4 °C in 2100). Model differences also cause a spread in the scenario outcomes[9]. Not only do models have parametric uncertainties in the estimations of processes such as technology learning rates[10], but model differences are also caused by fundamental structural differences[11,12] associated with the model type (for example, general versus partial equilibrium) or the role of cost optimality. Finally, apart from climate outcome and model characteristics, several scenario assumptions also influence the spread of energy futures. They range from socio-economic assumptions (for example, population and gross domestic product) to technological assumptions (for example, associated with hydrogen, bioenergy and carbon capture and storage (CCS)) and even scenario-specific narratives, normative descriptions or mechanisms (for example, changes in food consumption or trade patterns). Understanding the relative impact of these three drivers is important because it enables us to differentiate technologies and energy carriers, for which the projections primarily vary because of climate policy goals, from those that

[1]PBL Netherlands Environmental Assessment Agency, The Hague, The Netherlands. [2]Copernicus Institute of Sustainable Development, Utrecht Universiteit, Utrecht, The Netherlands. [3]National Institute for Public Health and the Environment, Bilthoven, The Netherlands. ✉e-mail: mark.dekker@pbl.nl

are determined mostly because of uncertainty resulting from model disagreement and scenario assumptions. For instance, this differentiation enables us to assess whether a reduction in fossil fuel use to reach the Paris goals is robust across models and scenarios. In other words, detecting the drivers of scenario spread enables us to assess the level of consensus on these projections.

Although there is growing attention to multi-model comparisons[13,14], previous work on consensus in mitigation strategies has been conducted mainly within closed diagnostic experiments with a confined selection of variables (for example, emission pathways or specific technologies such as bioenergy[13,14]), models[2], scenarios[15,16] or regions—complicating robustness analysis on a comprehensive scale and potentially yielding contradictions when comparing them. For example, emission projections in some ensembles are found to be most sensitive to model choice[17], whereas in other ensembles having a different focus or model set, emission projection variations across climate outcomes are found to supersede these model differences[18]. Moreover, although the assessment of the statistical significance of an outcome is common practice in multi-model studies[16,19], the quantification of the relative impacts of different drivers in determining the observed spread—which yields a more detailed perspective on agreement and uncertainty—is not. Another caveat in many scenario analyses is associated with a large bias towards a few high-abundance models: for example, 49% of the scenario entries in the AR6 database are produced by only two models. Therefore, in the current literature, a comprehensive and quantitative analysis of consensus and robustness of climate policy scenario outcomes is still pending, to our knowledge, despite the strong influence of these scenarios in IPCC reports.

We address this issue by identifying the cause of scenario spread—the climate target, the model used or scenario assumptions—across many aspects (variables) of the energy transition: greenhouse gas emissions, the total energy mix, the primary energy mix of electricity generation and the energy mix of end-use sectors (that is, the transportation, industry and buildings sectors). We use Sobol's method of variance decomposition[5] to discern the drivers of these variables—a method used commonly in sustainability research[20,21]. For example, to determine the impact of model differences on solar power projections, this method compares the overall variation in solar power to the extent to which it varies for each individual model. Although we can intuitively expect solar power to be high in scenarios with ambitious climate targets, we now provide insight into whether this can be concluded with statistical certainty from the available model projections or whether model and scenario assumption differences obscure this. Addressing the impact of scenario assumptions is crucial because it sheds light on the ongoing discussion on the sufficiency of current scenario differentiation[22–25]—that is, whether a wide range of (including less likely) energy futures and different assumptions on economic growth[26] are explored exhaustively. We revisit this topic later in the paper. An important contribution of our analysis is that we overcome the dominance of high-abundance models—a known problem in climate change mitigation literature[23,27]—using a debiasing procedure (Methods).

The analysis yields three indices reflecting the proportion of the spread explained by the three drivers (Methods). We indicate that there is high consensus that a particular variable is a robust element of mitigation policy if its value, for example, the level of emissions or energy carrier use, varies mostly across different levels of mitigation. Specifically, this is detected when the largest part of the total variation of the variable is found across climate targets, and therefore less across model differences and scenario assumptions.

## Electricity generation

In 2020, the power sector accounted for approximately 20% of the worldwide final energy consumption and 40% of the global $CO_2$ emissions[28]. A rapid shift in electricity generation is crucial to achieve the Paris Agreement climate goals: in many climate policy scenarios, electrification of end-use sectors is substantial and the power sector reaches net-zero emissions before other sectors[29]. Different technologies aid in achieving net-zero emissions: fossil fuel electricity generation plants can be equipped with CCS, replaced by intermittent renewable technologies such as solar and wind or replaced by nuclear, hydropower and biomass[30,31]. The use of biomass when combined with CCS can potentially lead to negative emissions because of the permanent sequestration of biogenic carbon[32]. The cost-optimal mix of technologies depends not only on assumptions about future costs and potential of these technologies but also on reliability and projections of energy demand and concomitant technology preferences.

We find that the phase-out of fossil fuels, use of (early-century total) renewables and the overall roll-out of CCS technologies in electricity generation are robust ingredients of climate mitigation strategies, whereas the relevance of individual technologies shows low model consensus. Figure 1a shows the quantification of these conclusions—the degree to which the spread observed in these variables is driven by climate outcome, model and other scenario assumptions. In Fig. 1a, we subdivide the energy sources in electricity generation into renewables, including biomass (green), sources involving CCS (blue) and fossil fuels without CCS (red). The interpretation of Fig. 1a is as follows. When an energy source is located in the top corner, climate targets mainly drive its variation. In other words, there is consensus that the level of ambition in mitigation scenarios has main implications for this energy source. By contrast, when the energy source is in the bottom left corner, the model used is the main determinant of the abundance of this energy source. The dimension of the other scenario assumptions (bottom right) reflects the portion of variance that is explained by scenario elements beyond the model, climate target or their interactions (see Methods for more details on the interpretation of this dimension).

The total use of renewables is robust only up to 2050 in mitigation strategies. Later in the century, most scenarios have high shares of renewables (Extended Data Fig. 7), but the exact value becomes more determined by model differences than the stringency of the climate target. Not only does the absolute renewable deployment differ substantially among models, but also its fraction of the total (Supplementary Information A.2). Other scenario assumptions have a limited impact on early-century renewables, partially reflected in a high second-order interaction term between the model and climate (Methods). The models agree on the qualitative importance of early-century renewables for mitigation goals, but differ on their timing (inertia) and magnitude of initial roll-out. The roll-out of individual renewable technologies is mainly varied by model differences even before 2050—the exception being wind power in 2030, where the climate target is the main driver. The gradually increasing model dependence for most renewable technologies can be explained by the expansion of volume differences of these renewables and the intensification of competition with other low-emission technologies such as CCS—which are sensitive to model differences on technology costs, potentials and energy demand. Electricity from biomass (without CCS), nuclear and hydropower is already mostly model dependent in 2030 (ref. 33). Although in most models nuclear power increases with climate ambition, there are large model differences in estimating costs and perspectives on nuclear risk factors[19,34]. Hydropower, being one of the cheaper renewable electricity technologies, is already close to its maximum potential in scenarios with limited mitigation, making the spread mainly driven by uncertainties around potentials[33] (Fig. 1c).

The spread in total use of energy sources that are combined with CCS is distinctly less determined by model differences than individual CCS technologies, and in 2030 and 2050 a large part of it is determined by other scenario assumptions. The variance in individual CCS technologies from coal, gas and biomass (BECCS) is driven by model differences

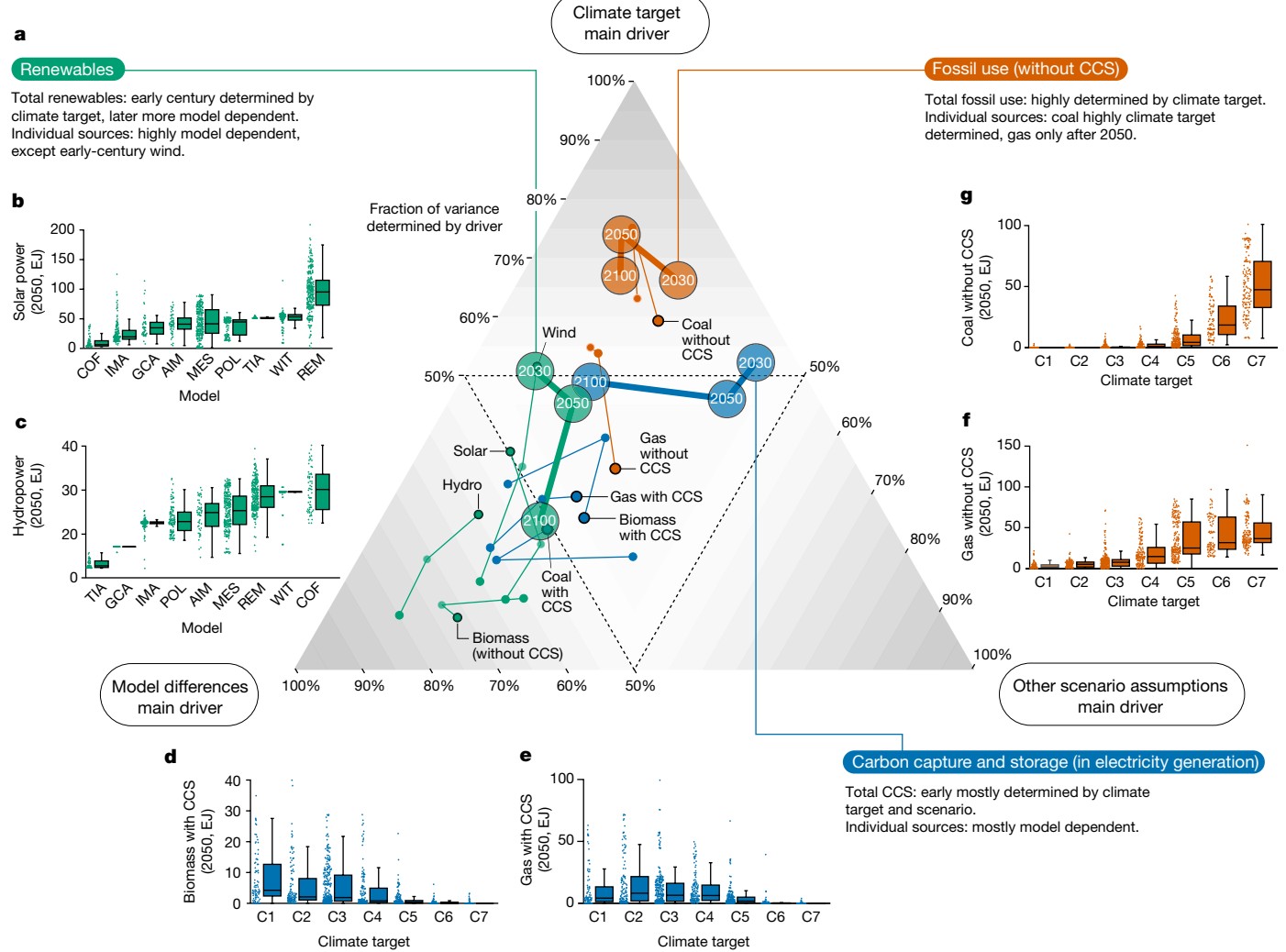

**Fig. 1 | Impact of model differences, climate targets and other scenario assumptions on the projections of electricity generation sources.**
**a**, Scenario spread in sources of electricity generation (secondary energy) is attributed to three drivers, depicted in the three corners: model differences, climate targets and other scenario assumptions. For example, fossil fuel use (red) being in the top corner indicates that the climate target dominates the scenario spread projected in fossil fuel use and that it is less affected by model differences and other scenario assumptions. Analogously, variables can instead be dominated by model differences (bottom left) or other scenario assumptions (bottom right). Dashed lines mark the 50% values in each of these three axes. The three main categories of sources of electricity generation are indicated using different colours: fossil fuels without CCS (red), sources involving CCS (blue), and renewables, including biomass (green). The totals in each category are shown with large dots marking the years 2030, 2050 and 2100.

Individual carriers or technologies in each category are shown with smaller dots and are annotated in the figure (in analogous colouring, and with 2030 shown using a black border). **b–g**, Projected 2050 values of individual sources of electricity generation, split by model (where models are identified by the three-letter abbreviations, defined in Extended Data Table 2) or climate category, based on the dominant driver (that is, we do not analyse other scenario assumptions in these subpanels): solar energy (**b**), hydroenergy (**c**), biomass with CCS (**d**), gas with CCS (**e**), gas without CCS (**f**) and coal without CCS (**g**). In box plots, centre line is the median, boxes the interquartile range, and whiskers the minimum and maximum values within ±1.5 times the interquartile range; a total of 1,152 individual scenario entries are shown in dots. The same figure using fractions of the total electricity generation (instead of absolute values) is shown in Extended Data Fig. 4.

associated with competition between CCS technologies. The level of BECCS in 2050 is also driven for a significant part by climate outcome, reflecting some consistency among models on its importance in reaching the climate goals. The slowly decaying climate dependence of the BECCS later in the century reflects the final BECCS use reaching maximum biomass potentials, yielding model dependency in the level of these potentials and suppressing additional rollout with increasing climate ambition (Fig. 1d).

In contrast to individual renewable and CCS technologies, the use of coal (Fig. 1g) and (late-century) gas (Fig. 1f) without CCS is strongly determined by the climate target. For climate outcomes of C3 (less than 2 °C temperature increase in 2100 with at least 67%

probability) and lower, coal use without CCS is phased out in practically all scenarios.

## Transport energy demand

The transport sector accounts for around 20% of global $CO_2$ emissions and 25% of total final energy consumption in 2020 (ref. 28). Like electricity, the transport sector plays a key part in reaching net-zero emissions[35], and different routes exist to decarbonize it[36,37]. Figure 2 shows the relative importance of the model, climate target and scenario differences on projections of energy sources for the transport sector. The primary finding is that the models disagree on early-century oil

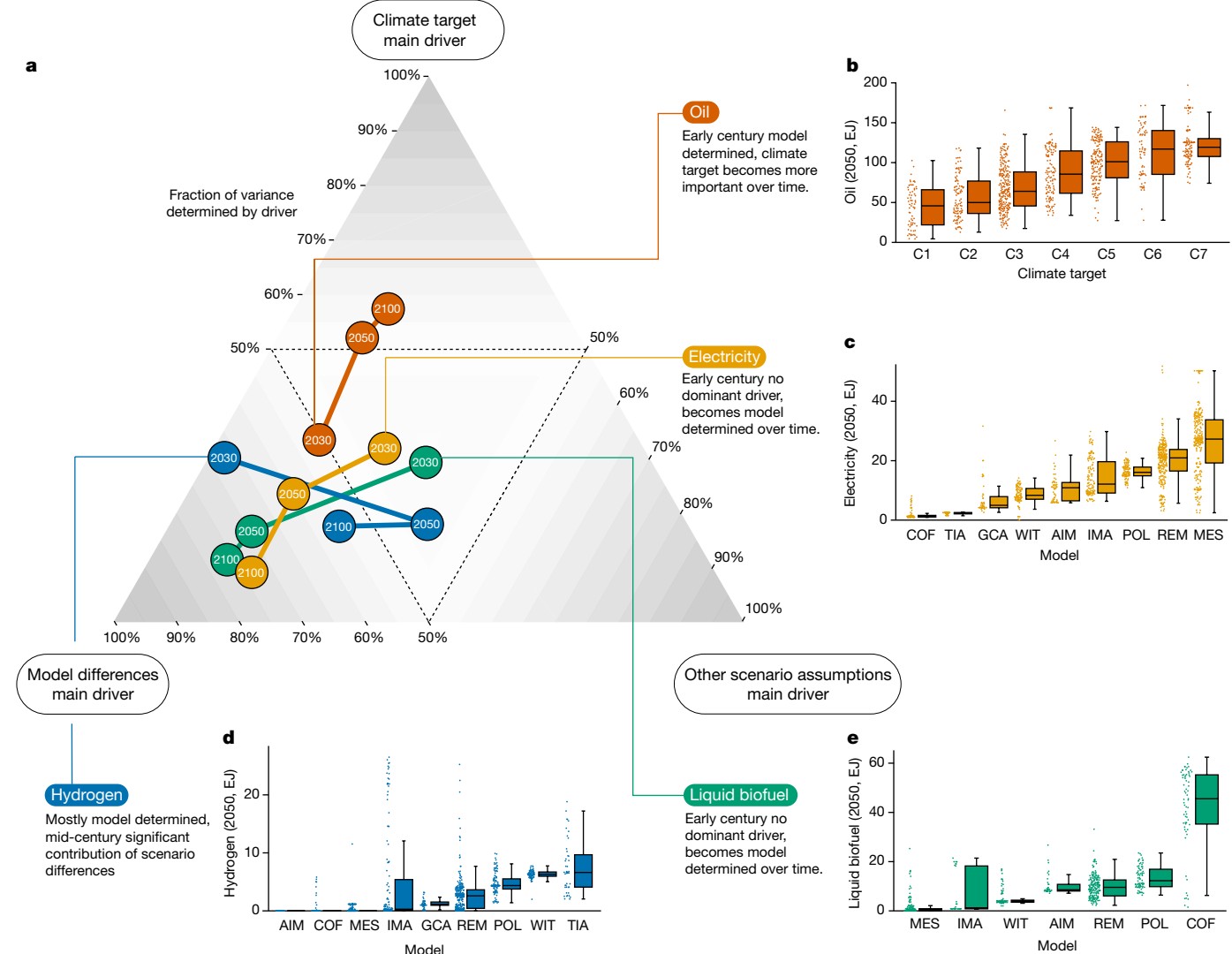

**Fig. 2 | Impact of model differences, climate targets and other scenario assumptions on the projections of energy sources in the transport sector.** **a**, Scenario spread of energy sources in the transport sector is attributed to three drivers, shown in the three corners: model differences, climate targets and other scenario assumptions. Spread in late-century oil (red) use in transport is mostly driven by climate targets (top corner), whereas other sources of energy in the transport sector (electricity in yellow, hydrogen in blue and liquid biofuel in green) are driven by model differences (bottom left corner), indicating the weak impact of other scenario assumptions (bottom right corner). Dashed lines mark the 50% values in each of these three axes. **b**–**e**, Projected 2050 values of the use of oil (**b**), electricity (**c**), hydrogen (**d**) and liquid biofuels (**e**) in the transport sector, split by climate target or model (for abbreviations, see Extended Data Table 2). In box plots, centre line is the median, boxes the interquartile range, and whiskers the minimum and maximum values within ±1.5 times the interquartile range; a total of 1,152 individual scenario entries are shown in dots.

use in transport, but over time become more aligned on the level of oil use in relation to the climate target. The use of electricity, liquid biofuels and hydrogen in transport is highly model dependent and, in contrast to oil, becomes more so over time. We discuss these findings in more detail below.

Oil (Fig. 2b) was by far the main energy carrier in transport in 2020, accounting for more than 90% of global final energy consumption[28]. After 2050, the level of oil use is mainly driven by climate outcomes. However, there is less consensus on how it is replaced. There are three main substitutes for oil in transport: electricity, hydrogen[28,38] and bioenergy.

Although electricity in the transport sector (Fig. 2c) has a substantial climate dependency early in the century, model differences are the main driver. This is especially true later in the century because in several models electrification of the transport sector also happens without stringent climate targets because of its projected increasing competitiveness (Extended Data Fig. 8). The reduced climate dependencies make the model differences relatively more important.

A similar conclusion can be drawn for hydrogen (Fig. 2d). Estimates of hydrogen use in transport vary greatly, as shown in Fig. 1e across different models in 2050. Observing the wide and for some models dichotomous distributions of hydrogen use in transport, this can partially be explained by the economics of scale for transport technologies[39]: either hydrogen rolls out substantially or it does so only very little and other technologies take over. These model differences are not superseded by differences across climate outcomes—even among the most ambitious scenarios (C1), several models project no hydrogen use at all in 2100 (Extended Data Fig. 9). Moreover, hydrogen use is sensitive to scenario assumptions, marked by the relatively rightward position of the 2050 value. Possible explanations are changes in the role of hydrogen in different versions of the same model and possible assumptions

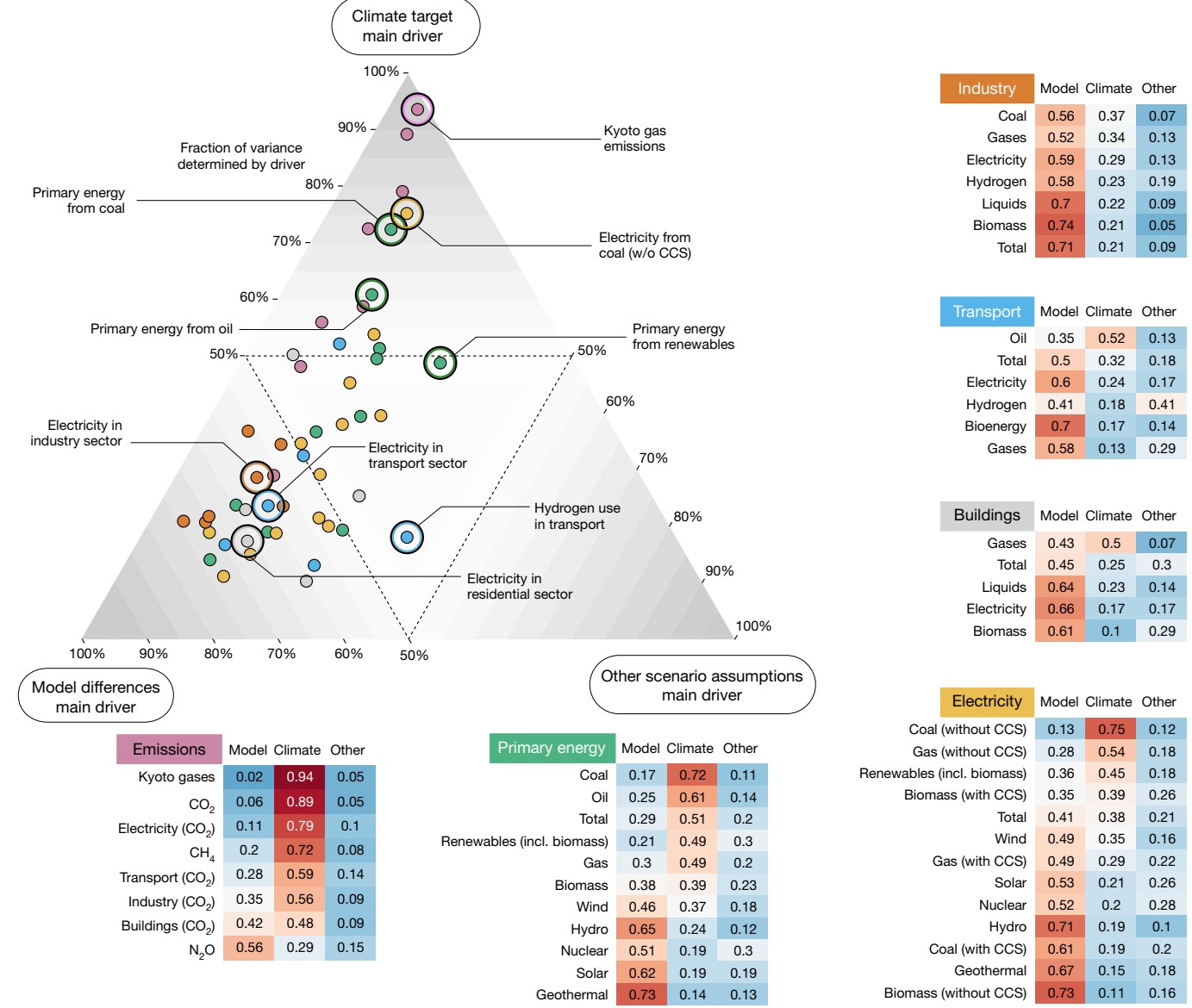

**Fig. 3 | Impact of model differences, climate targets and other scenario assumptions on the projections of the key variables in 2050.** Top left, an overview of the impact of the three drivers on 2050 projections of the key variables in the energy transition (for the assessment of 2100 projections, see Extended Data Fig. 3). The variables are split into six categories, identified by the colours corresponding to the tables to the side. The tables indicate the respective fractions of variance explained and are sorted by the variance explained by climate outcome differences. The values are highlighted in blue (low) and red (high) shades. The same figure using fractions of the total electricity generation (instead of absolute values) in the analysis is shown in Extended Data Figs. 5 and 6.

on the development of hydrogen and electricity infrastructure development.

Of the three main substitutes for oil in the transport sector, liquid biofuel (Fig. 2e) is the most model dependent in 2050. Comprising only 3.5% of the energy use in the transport sector at present[28], it starts in 2030 with a relatively low variance—also across models—that increases over time and moves far into the bottom left corner. Only seven out of nine models include bioenergy in transport, highlighting its model dependency, and one particular model has 4–40 times higher median bioenergy use in transport than other models in 2050, as observed in a previous work[40] (Extended Data Fig. 10). The strong influence of model structure on bioenergy deployment aligns with previous literature[13,14,40] and is also observed for other sectors (see below). These model discrepancies stem from the heterogeneous supply of bioenergy, encompassing various possible feedstocks, conversion routes and end-use possibilities.

## Overview of key variables in 2050

We extend the analysis to a larger group of key variables of the energy transition in Fig. 3, split into six categories: emissions (purple), primary energy (green), the industry sector (orange), the transport sector (blue), the building sector (grey) and electricity generation (yellow).

Distinct differences emerge among the six variable groups in Fig. 3. Climate outcomes predominantly shape emission variances. This is evident for Kyoto gases, $CO_2$ and even methane ($CH_4$). By contrast, nitrous oxide ($N_2O$) is more model dependent: although each model reduces $N_2O$ emissions in ambitious climate scenarios, disagreement over absolute levels outweighs this trend. Climate outcomes heavily influence sectoral emission levels, with the weakest signal in the buildings sector (48%). The use of most fossil fuels without CCS also shows a robust relation to climate outcomes, in addition to oil in transport;

gas in buildings; total use of renewables, including biomass; and, to a lesser extent, coal and gas in industry.

By contrast, the spread in individual renewable technologies and CCS technologies is mostly driven by model differences, although biomass shows no clear model or climate dependency. However, when looking at fractions rather than absolute values of the primary energy sources (Extended Data Fig. 4), biomass, wind and hydroenergy variations are, to a larger extent, driven by climate outcomes. The difference between the role of renewables in primary energy and the electricity mix is attributed to the reduction of the former in more ambitious scenarios (because of enhanced efficiency), even though the total electricity demand increases because of the electrification of end-use sectors. Although renewable energy use increases with climate ambition in both the primary energy and electricity mix, this trend is more strongly tied to climate ambition in primary energy use than in electricity generation.

Although end-use sector emissions and fossil fuel use are climate driven, the projection spread of non-fossil fuel energy use in these sectors is substantially affected by model differences. In particular, the spread in electricity use in all three sectors is explained by model differences for approximately 60% (Fig. 3), which is partly associated with differences in projecting the total energy consumption in these sectors[41]. However, the same analysis on electricity as a fraction of the total energy consumption per sector shows a much lower model dependency (down to approximately 40%; see Extended Data Fig. 6). More consensus (that is, climate dependency) is identified for the total electricity use as a fraction of the total final energy (not shown, peaking in 2050 with 53%), as well as the total final energy (49% in 2050).

Figure 3 demonstrates that variability in most quantities is primarily driven by climate outcomes or model differences. Other scenario assumptions have a varying impact but rarely dominate the variance. Examples in which they play a notable part include specific on–off assumptions in models (for example, hydrogen in transport and industry), lifestyle and policy-sensitive variables (for example, total energy consumption in the residential and commercial sectors or nuclear energy).

## Implications for policy and research

Based on the largest available set of climate policy scenarios[7], we provide a comprehensive overview of the drivers of spread in key mitigation variables in the IPCC AR6 database. In this way, we identify consensus as well as areas in which model differences—and biases—dominate the scenario projections. There are three merits of this paper.

The first merit is the quantitative overview of scenario variance drivers. It distinguishes scenario projections that show consensus across models and scenarios from those that are mostly dependent on model choice and scenario assumptions. This is relevant to researchers (by pointing out future research areas in which consensus is lacking and preventing scenario selection biases) as well as non-academic users of climate change mitigation scenarios. The identification of robust elements of climate change mitigation provides a foundation on which policymakers and stakeholders can make informed decisions and address key uncertainties. Some of the robust aspects of climate policy we find in this study are a quantitative confirmation of what is already known[42] or expected, such as the decrease in overall emissions as well as $CO_2$ emissions in individual end-use sectors and $CH_4$ emissions. Although the energy mix of end-use sectors shows large differences between models, the decrease in fossil energy use (mostly in transport and buildings) can also be regarded as robust. Furthermore, aggregated sets of technologies, such as early-century total renewable energy use, are found to be robust. To some extent, the variations in BECCS in the electricity mix, the fraction of wind, hydroenergy and biomass in the primary energy mix and electricity in end-use sectors

(as a fraction of the total energy use) are also primarily driven by climate targets. In smaller-scale studies, some of these variables are found to be still highly model dependent[2,43,44], which enables these findings to shed light on the robustness of their role in mitigation strategies.

However, apart from the exceptions mentioned above, a striking conclusion is the dominance of model differences in most (more specific) policy variables such as the deployment of individual technologies. Although this is in line with previous works on smaller-scale multi-model robustness of energy technology projections[16,19], this has not been quantified at the scale of this study and shown for many variables before. The causes of these model differences may not be similar for each variable. To this end, we refer to earlier works on model comparison[12,45,46] and more focused studies on, for example, bioenergy[40] and CCS[47]. Relating model differences to modelling pattern aspects has proved to be difficult[12,40].

We find a high model dependence of post-2030 solar energy (both primary and in the power mix), even though this is commonly argued to be of high importance to mitigation[48,49]. Individual CCS technologies in electricity generation are also highly model dependent, especially in coal and gas plants. Although the electricity fraction in demand sectors energy use is linked with climate targets, non-fossil energy sources at the end-use level are highly model dependent. Apart from detecting the dominant driver, the varying degrees of the three drivers provide a broader context: model disagreement, albeit dominant for many variables, is never explanatory for the full variable spread, and a non-negligible impact of climate targets can always be recognized (as is the case for, for example, CCS technologies and individual renewables). Although models tend to agree that bioenergy use expands from the current levels in mitigation scenarios, the supply and use of this resource differ considerably across models, particularly in the transport sector[36,50]. As highlighted in ref. 40, technology characterization and coverage vary markedly across models, particularly the technology deployment constraints. It is stressed that bioenergy deployment in mitigation scenarios is largely driven by the energy system context—for example, by the costs of alternative mitigation options.

The second merit is the scenario differentiation beyond the model or climate outcome. Although the impact of scenario assumptions is not uniform across variables (Fig. 3 and Methods), it is generally low. The insignificance of the variance explained by this dimension is a result in itself and raises questions about the representativeness of the AR6 database in covering a wide range of futures: partially, this is caused by most of the scenarios being based on the 'middle of the road' Shared Socioeconomic Pathway (SSP2) and cost-optimal assumptions. Although our methodology takes into account biases such as the varying amount of scenarios submitted by different models to the database, it does not correct for the overabundance of a particular scenario type such as SSP2. Theoretically, we expect a higher variety of outcomes in mitigation scenarios that do not mostly depend on climate targets or models, especially in the case of individual technologies, CCS and hydrogen that compete with each other but all contribute to mitigation. The limited variance driven by other scenario assumptions highlights the need for a systematic effort in representing diverse scenario assumptions. This scenario differentiation process requires a consistent exploration of normative assumptions and storylines, involving collaboration with various stakeholders, including policymakers and businesses. Important elements to explore include different economic growth rates (including post-growth scenarios[26]), globalization levels, technology preferences beyond costs, lifestyle changes and distinctions between technology-focused and sufficiency responses to climate and environmental challenges. A more comprehensive representation of different scenarios could enhance understanding of the driving forces determining energy futures, whereas model-based variability (Fig. 3, bottom left corner) primarily reflects a lack of consensus.

The third merit of this paper is shedding light on (the perception of) uncertainty and bringing a methodological advance by providing tools to be able to cope with it. The introduction of this method to the field of climate change mitigation will be useful for multi-model scenario studies and in future IPCC assessments to avoid incorrect perceptions of certainty. Our results have implications for how we perceive current literature on climate policy scenarios: we provide evidence that overall consensus on the roll-out of energy technologies and variables is generally lacking for all but fossil fuels and emissions. For non-modellers (such as policymakers), this change specifically addresses small-scale or single-model studies, as it is still relatively common in reports used for national and international policymaking—our results can act as a frame of reference for assessing the certainty of such projections, specifically on policy-relevant aspects that may previously be thought of as robust aspects of mitigation (for example, solar energy use). These studies are still common in science (for example, on solar energy[49], energy demand[51], energy access[52] and overall renewable energy[29]). However, not all model differences should be perceived as intrinsic uncertainties: some differences simply reflect different possible future outcomes that are equally consistent and not a result of unknowns or lack of understanding. In other words, model (and scenario) differences may identify the degrees of freedom of key mitigation variables (for example, for the specific energy mix), and a lack of them may point to a rather narrow space that policymakers have to navigate through (for example, for emission variables). Still, ideally, model differences are studied in rigorous scenario differentiation beyond individual models, which is lacking for most variables, as shown in Fig. 3.

The importance of model biases and the implicit lack of representation in scenario definitions point towards potential overinfluence and blind spots in different strategies to mitigate climate change in the available literature. At the same time, we also find that emissions, fossil fuel use and total renewables and CCS show robust values in mitigation, surpassing model differences so that they pinpoint areas of consensus. It is crucial for any user of mitigation scenarios—both academic and public—to be aware of these varying degrees of robustness in current mitigation literature. Apart from raising awareness, we believe designing alternative scenario narratives and using scenario-comparison methods that can detect bias and model impact should become a core part of future mitigation research.

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

## Methods

### General

We used Sobol's method of variance decomposition[5,53] in this analysis. By sampling datasets in which models and scenarios are uniformly represented, a measure of the intrinsic variance of each variable $v$ is obtained and decomposed into first- and second-order variance contributions of the aforementioned factors. This yields three indices for the variance explained by the climate category ($F_c$), model ($F_m$) and other scenario assumptions ($F_o$) that add up to 1. These indices are a combination of the traditional Sobol indices (as described below). If $F_c \rightarrow 1$, there is statistical consensus or robustness about $v$ being related to climate outcomes: for example, if although there may be strong model differences in projecting $v$, the differences between climate outcomes supersede that signal (that is, $F_c \gg F_m$). In this study, we treat these three indices as coordinates moving over time in a triangular variance decomposition landscape (Figs. 1–3), showing how the variance of each variable is determined, whether there is statistical consensus about its relation to climate outcomes and how this changes over time in our projections of the next century.

In this paper, we decompose the variance of energy variables by using the actual values of each variable. Whether we decompose the variance of the absolute values of the variable or instead the variance of its fraction of the total energy consumption can strongly affect the results, yielding a richer interpretation if both are analysed. In the main text, we have chosen to show only the decomposition results of the variance of the absolute values of each variable because the fractional counterpart is not fully defined or of interest for all variables (for example, for emission variables). It is more intuitive and consistent to take a single approach for all the variables. Nevertheless, both the absolute and fractional values of many energy variables can be policy relevant, and insights from both are therefore used in the discussion. The results of the same analysis on the fractional values of several variables are shown in Supplementary Information A.2.

### Database

The AR6 database[7] is a product of the IPCC AR6 WGIII report on the Mitigation of Climate Change[1]. In this analysis, we focus on the scenarios that passed the historical vetting. In this way we exclude scenarios with historical values that are much different from the observations and use only those that have a climate assessment of the resulting emissions. This yields a subset of 1,202 scenarios, with 44 unique model versions, 13 unique model frameworks and 8 different climate categories. The climate categories are shown in Extended Data Table 1.

Extended Data Table 2 shows the models in the global version of the database that make projections on the key variables assessed in this paper. The table also indicates that we aggregate model versions onto single-model labels; for example, both IMAGE 3.0 and IMAGE 3.2 are identified as IMAGE (see section 'Pre-processing').

Therefore, the models in the database (Extended Data Table 2) do not have equal numbers of entries across all climate categories (Extended Data Table 1). As an example, Extended Data Table 3 provides the number of historically vetted scenarios per model and climate category for primary energy from coal. Orange cells have been removed from the dataset because of the low abundance of C8, EPPA and MERGE-ETL scenarios: including them would make well-representative samples impossible. Moreover, the empty entries in the blue cells for TIAM-ECN, which are included, make the sampling method slightly imperfect. However, because it concerns only 2 out of 63 entries, the term is still small and the interpretation of the indices is approximately the same. An alternative would have been to fully drop the TIAM-ECN model as well, which in turn also decreases the representativeness of the sample. Hence, we have chosen to keep this model in and accept these empty entries.

### Variance decomposition

For each variable $v$, a decomposition of the variance is performed at each time step $t$. In particular, we use Sobol's method, which is based on the reasoning that the value of a variable can be written as a function $f(x_1, x_2, ..., x_n)$ of several independent inputs $x_i$. In our case, we have two identifiable inputs: the climate target $x_c$ and the model $x_m$ used to calculate the variable. Because that does not cover all variation that we observe in the dataset and we do not have other information distinguishing the scenario entries apart from their model and climate targets, we add a noise term $\zeta$, leading to the following expression of the so-called Hoeffding–Sobol decomposition applied to our case:

$$v(t) = f(t, x_c, x_m)$$
$$= f_0(t) + f_c(t, x_c) + f_m(t, x_m) + f_{cm}(t, x_c, x_m) + \zeta(t, x_c, x_m)$$

where the first term on the right-hand side is an overall average, $f_i$ is a function of only factor $i$ and $f_{cm}$ is a function of both $x_m$ and $x_c$. The final (noise) term is also dependent on $x_m$ and $x_c$: we do not know a priori whether this noise is independent of the climate target and/or model. Following the line of Sobol's theory, this noise term contains both the first-order impact of other scenario assumptions on $v$ and potential second- and even third-order terms between these assumptions with climate targets and models. Although $v(t)$ is inherently a function of time, the variance decomposition is done for individual moments in time $t_0$. For clarity, in the sequel, we therefore drop the term $t$ in the equations.

Although $f_0$ is merely the overall average of $v$, the higher-order functions $f_x$ are expressed as conditional expected values: for example, $f_m = E(v|x_m) - f_0$ and $f_{mc} = E(v|x_m, x_c) - f_0 - f_c - f_m$. Taking the square integral of the above equation over $\mathbf{x}$ and dividing by the total variance of $v$ ultimately yields

$$1 = \frac{\text{var}(f_c(x_c))}{\text{var}(v)} + \frac{\text{var}(f_m(x_m))}{\text{var}(v)} + \frac{\text{var}(f_{cm}(x_c, x_m))}{\text{var}(v)} + \frac{\text{var}(\zeta(x_c, x_m))}{\text{var}(v)}$$

The last term can be interpreted as the total variance (including both first- and higher-order terms) explained by scenario assumptions other than climate target and model choice. Because it is not an actual first- or second-order term, we write it as $S'_o$. For the other terms, we use the definitions of Sobol indices $S_i = \text{var}(f_i)/\text{var}(v)$, yielding

$$S_c(v) + S_m(v) + S_{cm}(v) + S'_o(v) = 1$$

The inputs $x_c$ and $x_m$ are, if taken from the dataset directly, not independent: for example, some models have much more entries for C4 than for C2, whereas other models have the opposite. This makes the Hoeffding–Sobol decomposition invalid: the terms $S_c$ and $S_m$ would partly cover the same variance because of covarying labels. We solve this by not determining the variances directly from the database entries, but by creating sampled datasets such that all climate categories and models are uniformly represented, and in turn apply the variance decomposition on these sets. In practice, this works as follows. For each model–climate category pair (for example, REMIND–C1; Extended Data Table 3), we draw $p_{\text{sample size}} = 3,000$ scenarios. Because the combination REMIND–C1 does not have 3,000 scenarios, we draw these samples by allowing replacement of the draws. After doing this for all climate–model pairs, we obtain a large dataset in which $x_c$ and $x_m$ are perfectly orthogonal. From this set, we calculate the Sobol indices. Because this process is stochastic, we redo this process ($p_{\text{resample}} = 100$ times) and report the average. This process involves two parameters: $p_{\text{sample size}}$ and $p_{\text{resample}}$. If these parameters are taken too small, the results may be prone to stochasticity and the sample may lack sufficient uniformity, yielding errors in the indices. The values of 3,000 and 100, respectively, are found to be high enough and approximating a deterministic

result: when performing the same analysis using $p_{\text{sample size}} = 1,000$ and $p_{\text{resample}} = 30$, the values of the indices $F_c$, $F_m$ and $F_o$ (defined below) changed on average with only 0.0007, 0.0008 and 0.0009, respectively, in 2050 (see Methods and Extended Data Fig. 1 for details on parameter sensitivity). In a previous work[53], the calculation of the Sobol indices has been rewritten in matrix form, which is also what we use in favour of computational efficiency.

The sampling method removes bias stemming from differences in the abundance of models. The method assumes that the scenario entries per climate category and model are representative of these labels, which arguably holds better for model–label combinations with many entries, than for those with just a few. TIAM-ECN does not have C1 and C2 entries, meaning that these entries are empty in the sampling, and the sample is not perfect.

We aim to decompose the total variance into terms attributable to each individual input. However, in contrast to a similar approach in an earlier work[2], the second-order Sobol term $S_{cm}$ cannot be neglected, as for most variables $v$: $0.05 < S_{cm}(v, t = 2050) < 0.30$ (average 0.16). From an intuitive point of view, the second-order variations also matter: when all models show variation among climate targets in their output, but in different magnitudes or at a different base level–so that it is less pronounced in the first-order term–this is of interest. Moreover, if the second-order term would be excluded, we cannot interpret the indices as fractions of the total variance anymore, as they do no longer add up to 100%. For these reasons, we construct three indices based on the calculated first- and higher-order Sobol terms, in which we add $S_{cm}$ to $S_c$ and $S_m$:

$$
\left\{
\begin{array}{c}
F_c := S_c + \dfrac{1}{2} S_{cm} \\[4pt]
F_m := S_m + \dfrac{1}{2} S_{cm} \\[4pt]
F_o := S'_o = S_o + S_{om} + S_{oc} + S_{ocm}
\end{array}
\right.
$$

which add up to 1. The indices $S_o$, $S_{om}$, $S_{oc}$ and $S_{ocm}$ are mentioned here for interpretation purposes but cannot be explicitly calculated in the analysis: together, they form a 'rest' term and do not govern a defined set of assumptions. We cannot calculate $F_o$ directly because we lack appropriate labels on other scenario assumptions, and moreover, we lack enough, well-distributed data to create a sample from which we can explicitly compute these individual terms. Therefore, we deduce $F_o$ from $F_c$ and $F_m$. Note that the expression for $F_o$ also contains terms with 'model' and 'climate' subscripts. Although it is not possible to compute them separately with the available data, we think adding them together is appropriate as long as the interpretation is clear: $F_o$ not only takes into account model- and climate-ambition-independent variations, for example, because of assumption differences on gross domestic product or technological advancement, but also takes into account how these assumption differences vary across models and climate ambition–varying gross domestic product may have a different or more pronounced effect on model X than it has on model Y. However, for the identification of robust aspects, these model differences are of less interest, which legitimizes the choice of grouping them into $F_o$. Thus, for both practical reasons and interpretation reasons, we have chosen these definitions of the indices.

This means that the total fraction of variance explained by model (climate) differences may therefore be slightly higher than $F_m$ ($F_c$). This way, $F_o$ acts as an upper bound on the relative effect of other scenario assumptions, both in lower and in higher orders. However, because the relative magnitude of $F_o$ turned out to be so small and that other scenario assumptions are not systematic among model entries (climate outcomes) we expect that the total sensitivity towards model (climate) differences is already approximated by $F_m$ ($F_c$)–although the mathematical definition of $F_m$ ($F_c$) includes only the first-order and second-order terms between the model and climate.

In summary, the interpretation of the resulting indices $F_i(v, t)$ is the percentage of the variance of variable $v$ in year $t$ that is explained because of differences in factor $i$ (that is, model, climate target or other scenario assumptions), where all effects of other scenario assumptions are aggregated in the last index. All higher-order terms are accounted for.

## Sensitivity of the decomposition results

As mentioned earlier, there are two parameters important to the analysis: the sample size ($p_{\text{sample size}}$) and the number of times that the analysis is performed again ($p_{\text{resample}}$) before averaging the results. The latter parameter modulates the number of times the samples are redrawn and the variance decomposition is performed again. In the main results, we use $p_{\text{sample size}} = 3,000$ and $p_{\text{resample}} = 100$. Note that $p_{\text{resample}}$ is larger than 1 to average out potential stochasticity that is inherent in a non-infinitely-sized sample drawing. The sensitivity of model results is illustrated by varying $p_{\text{sample size}}$ and reporting the $F_m$ for solar power. Using $p_{\text{resample}} = 100$, we can identify the convergence of the results on increasing $p_{\text{sample size}}$. These results are shown in Extended Data Fig. 1. Averaging the results from a larger number of redraws is important up to $p_{\text{sample size}} = 1,000$, after which the confidence intervals become narrow and each individual sample becomes very close to the average. Averaging over 100 samples then results in a near-constant value.

## Second-order model–climate interaction term

Extended Data Table 4 shows the second-order model–climate interaction term ($S_{mc}$), as well as the multi-order coefficient used to reflect the fraction of variance determined by other scenario assumptions ($F_o$), for the sources of electricity generation–similar to the variables in Fig. 1 of the main text. Extended Data Table 4 shows that $S_{mc}$ cannot be neglected, which in the past was possible in similar efforts, but using a more confined ensemble. For CCS, these coefficients range between 18% and 28% for individual sources (coal, gas and biomass with CCS), whereas the total is lower. For fossil use and renewables, however, these coefficients vary over time. The model–climate interaction terms (for both total and individual sources) for fossils grow substantially over time, reaching 25–35% of the variance explained. The late-century distribution of fossils over the climate categories differs per model: for example, there is discrepancy among models on the exact temperature outcome for which gas in 2100 should be reduced to zero–more so than coal.

Generally, renewables (both the total and individual sources) have a rather large model–climate interaction term in 2030, which drops in later years. Extended Data Fig. 2 shows the spread of renewable energy use for electricity generation in 2030. It is visible that in 2030, the differences between climate outcomes (top right) are not yet well pronounced; only for C1 the renewables are significantly more abundant. Model differences (top left) are already distinguishable, but the distributions still contain a lot of overlap. Splitting across both models and climate outcomes (bottom), we can see that the models estimate the relationship between the renewables and climate outcomes rather differently (that is, resulting in a high value of $S_{cm}$): for WITCH, the relation with C categories is almost non-existent (that is, horizontal), whereas for AIM, it is approximately linear and for IMAGE or MESSAGE, an exponential relation is visible. In other words, the short-term (2030) response to carbon taxes (either in terms of timing or overall magnitude), as opposed to how renewables develop in the baseline, is distinctly different across models, resulting in a high value of $S_{mc}$.

## Dimensions of other scenario assumptions

In Extended Data Table 4 (right), the metric we use for the other scenario assumptions ($F_o$) is shown for the different sources of electricity generation. For reference, we repeat the definition of $F_o$:

$$F_o := S_o + S_{om} + S_{oc} + S_{ocm}$$

As seen in this equation, this term includes not only the first-order effect of (not explicitly labelled) scenario assumptions ($S_o$) but also the higher-order terms for which those other scenario assumptions interact with either climate outcome or model choice ($S_{om}$, $S_{oc}$ and $S_{ocm}$). Extended Data Table 4 shows that the term is not negligible—it is neither always low nor uniform across the variables in the table, spanning between 6% and 41% of the total variance just for this set of variables. Because this term inversely shows how important the model and climate outcome dimensions are, it is crucial to be aware of its value.

Apart from indicating the importance of other dimensions, the value of $F_o$ provides lessons about scenario differentiation. For example, Extended Data Table 4 shows that the 2030 values of $F_o$ for fossil fuel (without CCS) and CCS sources are relatively high. The value of 39% (41% even in 2050) for the total CCS sources in electricity generation is notable. There is a substantial early-century scenario variation in these variables, showing a wide spread of possible values of these variables, projected by the same model and within the same climate outcome. In the late century, this changes for most variables, indicating that different scenarios of the same model and with the same climate outcome seem to converge to (relatively more) similar values of CCS and fossil use. (Note that $F_o$ is a relative metric: it could be that the scenario differences are of similar magnitude, but become relatively smaller over time because differences between models and between climate outcomes become more expressed over time.)

## Pre-processing

Several pre-processing steps are implemented before the variance decomposition. The first step is cleaning the data by removing a few outliers. In particular, two scenarios (EN_NPi2020_800 and EN_NPi2020_900 by WITCH 5.0) are removed because they showed unrealistically high values for hydrogen use in transport (of the order of 1,000 exajoules, whereas the total energy use in transport was a factor 10 lower and most other entries of hydrogen in transport are a factor 100 or even 1,000 lower).

The next step is aggregating the model versions into single models. For example, we do not distinguish IMAGE 3.0 from IMAGE 3.2. The reason is that if we would keep them separate, it would result in an unrealistic model similarity: the model category would explain the sudden decrease in variance because the models seem to be more similar to each other (that is, $S_m$ drops sharply), although in reality this is not caused by unique models, but by unique model versions. Note that some models have more than 10 versions of themselves reported as unique model versions. In this paper, we are interested in how different modelling perspectives or modelling groups project variables differently. We believe that this is best illustrated when distinguishing model frameworks from each other rather than mere model versions. Extended Data Table 2 describes the exact translations between models and model versions. Model versions may still differ, which are now not recognized as model differences, but as differences among scenarios of the same model group, mostly contributing to $F_o$.

Scenarios with climate category C8 (exceed warming of 4 °C by 2100 with more than 50% probability) are removed, because only four out of nine models (that in general have sufficient entries) report C8 scenarios, leaving a model bias in how C8 is reported. Note that these models cover a small fraction of the total set anyway (about 2% for most variables). After this pre-processing step, 1,152 scenarios remain.

It is important to note that not all scenarios contain the same sets of variables. Some more detailed variables (for example, hydrogen use as fuel for specifically passenger transport) are covered by only a few hundreds of scenarios. To keep the number of scenarios used in the analysis of each variable as high as possible, we create separate databases for each unique variable $v$, containing all scenarios (out of the aforementioned 1,152) that contain $v$. In each database, we remove all scenario entries of models that have less than 10 entries in total. In practice, the results are not sensitive to the value of this parameter for a broad range of its values because a clear separation of small-abundance models can already be recognized (Extended Data Table 3), the numbers being 1, 7, 45, 47, 55, 65, 113, 114, 142, 266 and 297. The models having only one (MERGE-ETL) and seven (EPPA) scenario entries are much lower than the rest and are therefore removed in this analysis—because we also aim to distinguish climate categories within the model entries.

In most of the scenarios, data is provided in 5-year increments. For some scenarios, however, data may be missing for certain time steps or only 10-year increments are reported in the second half of the century. For this study, the temporal resolution must be fully equal among the scenarios, which is why we fill in these gaps using linear interpolation within the scenario entry such that all scenario entries have 5-year increments.

## Limitations

There are several limitations of this methodology, some of which are already mentioned before. First, the number of entries with combined model and climate category labels varies greatly (Extended Data Table 3). This was the reason behind applying the sampling method but raises questions on the respective representativeness of each model–climate combination. For example, there is only one single COFFEE-C1 scenario: the question is whether this single scenario represents this combined label enough. Arguably, the 84 REMIND-C3 scenarios are a better representation of their combined label. Second, other scenario assumptions (beyond climate outcome) are not systematically varied along models and climate categories, making the sampling for this not accurately interpretable. In future research, similar analyses could be performed on databases in which the SSPs are well represented among all model–climate combinations. Third, as already mentioned, other scenario assumptions may have nonzero higher-order terms involving climate and model differences, as well, which is taken into account but cannot be explicitly taken apart from the other terms in $F_o$. Finally, the sampling method takes care of bias towards high-abundance models and climate outcomes but is limited by the database itself: potential biases in the full scientific integrated assessment modelling community cannot be filtered, which are potentially not negligible.

A few final considerations about the term 'consensus' should be noted. In this paper, we refer to consensus about a link of the variable to mitigation strategies when its variance is mainly driven by climate outcome and less so by other drivers. However, this does not necessarily mean that this variable has no uncertainty or spread anymore: it can very well be that the exact value of the respective variable still covers a broad range even when considering a single climate target. Also, when a variable is significantly driven by other scenario assumptions (bottom right corner of the triangular panels in the figures), other forms of consensus may be present. For example, when there is an agreement (consensus) on a plural set of energy futures, depending on a set of scenario assumptions, that are all possible under similar climate outcomes and similar models. This type of consensus can be an interesting future research avenue, also in light of the finding that only very little (relative) variance is captured by scenario assumptions.

## Data availability

The input data for this analysis are obtained from the IPCC Sixth Assessment Report (AR6) scenario database[7] v.1.1. This database is an open source and can be accessed at IIASA (data.ece.iiasa.ac.at/ar6). The output data files contain processed scenario output and the fractions of variance explained by each driver, for a large variety of variables. These files are published on Zenodo and can be accessed publicly at https://doi.org/10.5281/zenodo.8221035. The source data for all figures can also be found in the Zenodo repository.

## Code availability

The source code for handling the data, applying the decomposition and creating the figures in both the main text and Supplementary Information can be publicly accessed on Zenodo (https://doi.org/10.5281/zenodo.8221035) and is maintained on GitHub (https://github.com/MarkMDekker/ar6_variance_decomposition). The code is written in Python v.3.9.16.

53. Saltelli, A. Making best use of model evaluations to compute sensitivity indices. *Comput. Phys. Commun.* **145**, 280–297 (2002).

**Acknowledgements** This work was supported by the European Climate and Energy Modelling Forum (ECEMF, H2020 grant agreement no. 101022622), the Next Generation of Advanced Integrated Assessment Modelling to Support Climate Policy Making (NAVIGATE, H2020 grant agreement no. 821124) and the Exploring National and Global Actions to reduce Greenhouse gas Emissions (ENGAGE, H2020 grant agreement number 821471).

**Author contributions** All authors conceived the study. M.M.D. performed the analysis, generated the figures and wrote the first draft. M.M.D., K.v.d.W. and R.v.H. contributed to the methods. A.F.H., M.v.d.B, D.P.v.V., V.D. and M.M.D. analysed the output. All authors contributed to the writing of the paper.

**Competing interests** The authors declare no competing interests.

**Additional information**
**Correspondence and requests for materials** should be addressed to Mark M. Dekker.

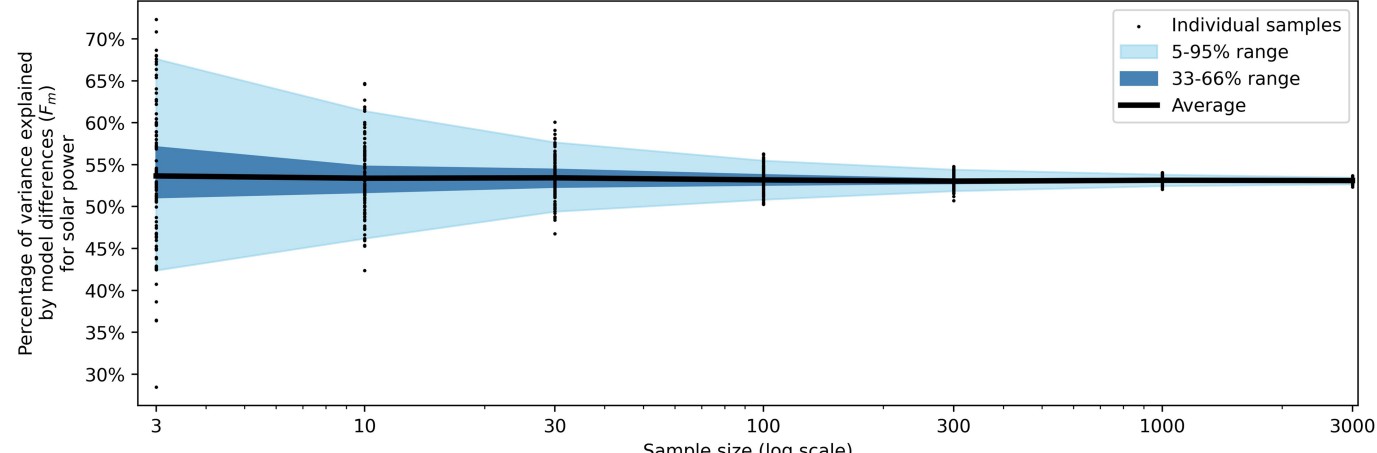

**Extended Data Fig. 1 | Sensitivity and convergence of results based on parameter settings.** Fraction of solar power variance driven by model differences ($F_m$) for different values of $p_{\text{sample size}}$ – i.e., the number of scenario-climate category pairs drawn in each sample (per unique scenario-climate category combination). Confidence intervals show the range of values reached with resampling 100 times (individual samples are shown in dots). In the main results of this paper, we also resample ($p_{\text{resample}} = 100$) times and take the average, and we use $p_{\text{sample size}} = 3{,}000$.

Secondary Energy|Electricity|Renewables (incl. Biomass) in 2030

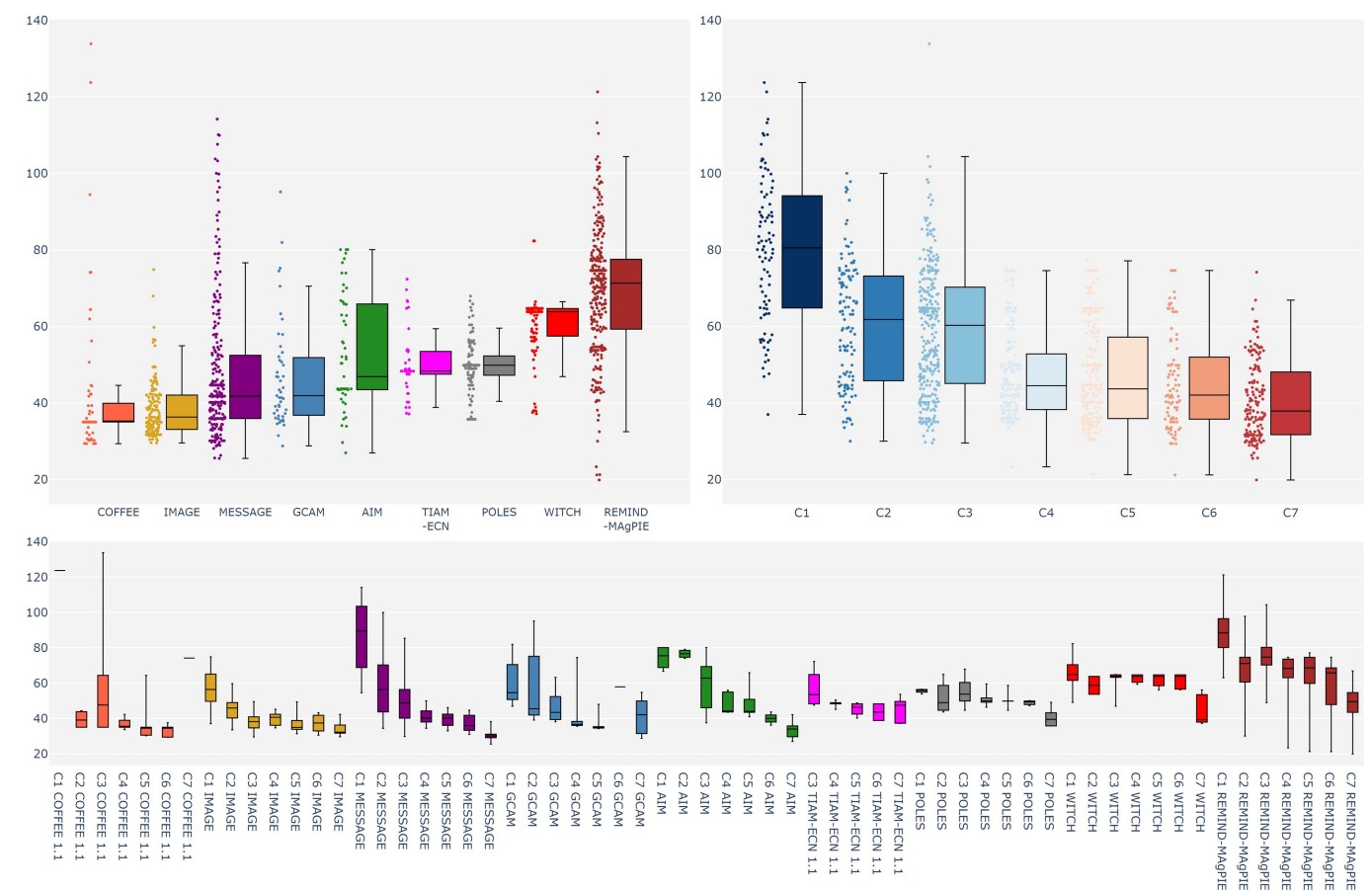

**Extended Data Fig. 2 | Projections of renewables (including) biomass in 2030, shown per model and climate category.** Data of renewables (including biomass) used for electricity generation in 2030, sorted by model (**upper-left**), climate category (**upper-right**) and both (**bottom**). Boxplots indicate quartiles (excluding outliers) and each dot reflects a single scenario projection.

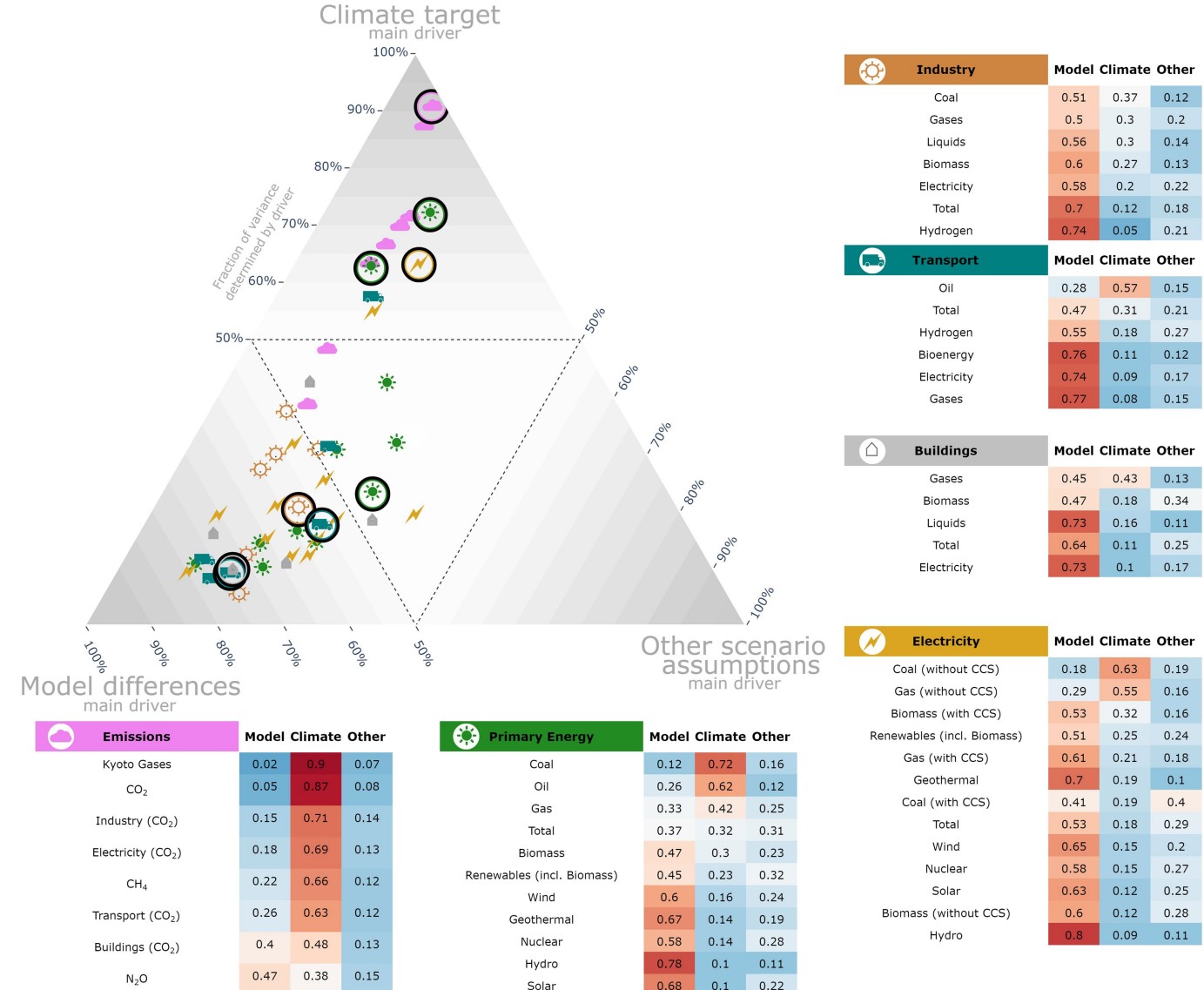

| Industry | Model | Climate | Other |
| --- | --- | --- | --- |
| Coal | 0.51 | 0.37 | 0.12 |
| Gases | 0.5 | 0.3 | 0.2 |
| Liquids | 0.56 | 0.3 | 0.14 |
| Biomass | 0.6 | 0.27 | 0.13 |
| Electricity | 0.58 | 0.2 | 0.22 |
| Total | 0.7 | 0.12 | 0.18 |
| Hydrogen | 0.74 | 0.05 | 0.21 |

| Transport | Model | Climate | Other |
| --- | --- | --- | --- |
| Oil | 0.28 | 0.57 | 0.15 |
| Total | 0.47 | 0.31 | 0.21 |
| Hydrogen | 0.55 | 0.18 | 0.27 |
| Bioenergy | 0.76 | 0.11 | 0.12 |
| Electricity | 0.74 | 0.09 | 0.17 |
| Gases | 0.77 | 0.08 | 0.15 |

| Buildings | Model | Climate | Other |
| --- | --- | --- | --- |
| Gases | 0.45 | 0.43 | 0.13 |
| Biomass | 0.47 | 0.18 | 0.34 |
| Liquids | 0.73 | 0.16 | 0.11 |
| Total | 0.64 | 0.11 | 0.25 |
| Electricity | 0.73 | 0.1 | 0.17 |

| Electricity | Model | Climate | Other |
| --- | --- | --- | --- |
| Coal (without CCS) | 0.18 | 0.63 | 0.19 |
| Gas (without CCS) | 0.29 | 0.55 | 0.16 |
| Biomass (with CCS) | 0.53 | 0.32 | 0.16 |
| Renewables (incl. Biomass) | 0.51 | 0.25 | 0.24 |
| Gas (with CCS) | 0.61 | 0.21 | 0.18 |
| Geothermal | 0.7 | 0.19 | 0.1 |
| Coal (with CCS) | 0.41 | 0.19 | 0.4 |
| Total | 0.53 | 0.18 | 0.29 |
| Wind | 0.65 | 0.15 | 0.2 |
| Nuclear | 0.58 | 0.15 | 0.27 |
| Solar | 0.63 | 0.12 | 0.25 |
| Biomass (without CCS) | 0.6 | 0.12 | 0.28 |
| Hydro | 0.8 | 0.09 | 0.11 |

| Emissions | Model | Climate | Other |
| --- | --- | --- | --- |
| Kyoto Gases | 0.02 | 0.9 | 0.07 |
| CO$_2$ | 0.05 | 0.87 | 0.08 |
| Industry (CO$_2$) | 0.15 | 0.71 | 0.14 |
| Electricity (CO$_2$) | 0.18 | 0.69 | 0.13 |
| CH$_4$ | 0.22 | 0.66 | 0.12 |
| Transport (CO$_2$) | 0.26 | 0.63 | 0.12 |
| Buildings (CO$_2$) | 0.4 | 0.48 | 0.13 |
| N$_2$O | 0.47 | 0.38 | 0.15 |

| Primary Energy | Model | Climate | Other |
| --- | --- | --- | --- |
| Coal | 0.12 | 0.72 | 0.16 |
| Oil | 0.26 | 0.62 | 0.12 |
| Gas | 0.33 | 0.42 | 0.25 |
| Total | 0.37 | 0.32 | 0.31 |
| Biomass | 0.47 | 0.3 | 0.23 |
| Renewables (incl. Biomass) | 0.45 | 0.23 | 0.32 |
| Wind | 0.6 | 0.16 | 0.24 |
| Geothermal | 0.67 | 0.14 | 0.19 |
| Nuclear | 0.58 | 0.14 | 0.28 |
| Hydro | 0.78 | 0.1 | 0.11 |
| Solar | 0.68 | 0.1 | 0.22 |

**Extended Data Fig. 3 | Impact of model differences, climate targets or other scenario assumptions on the projections of many key variables in the year 2100.** As in Fig. 3 in the main text, but for the year 2100 rather than 2050. Highlighted variables in upper-left panel are those that are highlighted in Fig. 3 as well.

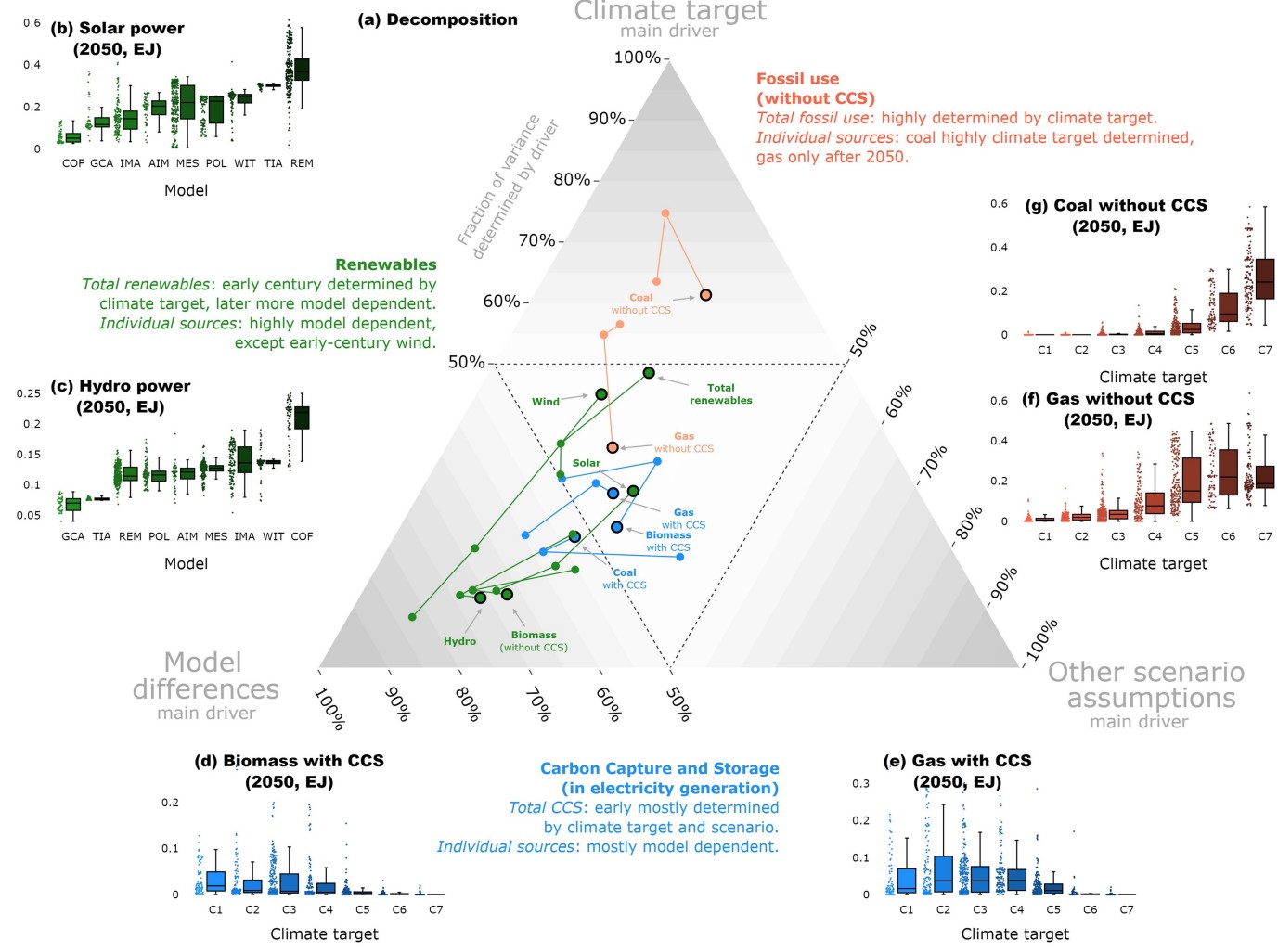

**Extended Data Fig. 4 | Impact of model differences, climate targets or other scenario assumptions on the projections of electricity generation sources as fractions of the total electricity generation.** Variance decomposition results for energy mix used for electricity generation similar to Fig. 1 of the main text, but not using the absolute value of these variables (in exajoules), but their fraction of the total energy used. Subpanels (b)-(e) are also changed accordingly.

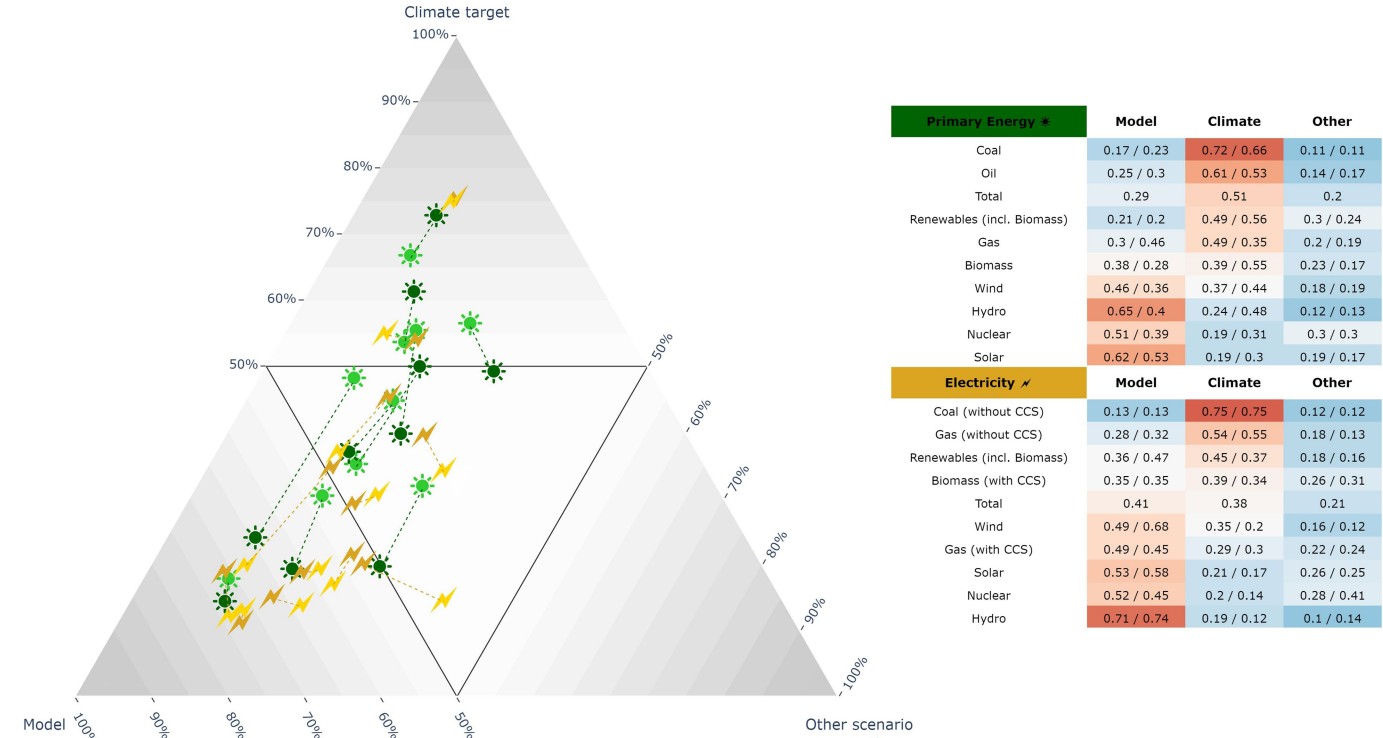

| Primary Energy ☀ | Model | Climate | Other |
|---|---|---|---|
| Coal | 0.17 / 0.23 | 0.72 / 0.66 | 0.11 / 0.11 |
| Oil | 0.25 / 0.3 | 0.61 / 0.53 | 0.14 / 0.17 |
| Total | 0.29 | 0.51 | 0.2 |
| Renewables (incl. Biomass) | 0.21 / 0.2 | 0.49 / 0.56 | 0.3 / 0.24 |
| Gas | 0.3 / 0.46 | 0.49 / 0.35 | 0.2 / 0.19 |
| Biomass | 0.38 / 0.28 | 0.39 / 0.55 | 0.23 / 0.17 |
| Wind | 0.46 / 0.36 | 0.37 / 0.44 | 0.18 / 0.19 |
| Hydro | 0.65 / 0.4 | 0.24 / 0.48 | 0.12 / 0.13 |
| Nuclear | 0.51 / 0.39 | 0.19 / 0.31 | 0.3 / 0.3 |
| Solar | 0.62 / 0.53 | 0.19 / 0.3 | 0.19 / 0.17 |
| Electricity ⚡ | Model | Climate | Other |
| Coal (without CCS) | 0.13 / 0.13 | 0.75 / 0.75 | 0.12 / 0.12 |
| Gas (without CCS) | 0.28 / 0.32 | 0.54 / 0.55 | 0.18 / 0.13 |
| Renewables (incl. Biomass) | 0.36 / 0.47 | 0.45 / 0.37 | 0.18 / 0.16 |
| Biomass (with CCS) | 0.35 / 0.35 | 0.39 / 0.34 | 0.26 / 0.31 |
| Total | 0.41 | 0.38 | 0.21 |
| Wind | 0.49 / 0.68 | 0.35 / 0.2 | 0.16 / 0.12 |
| Gas (with CCS) | 0.49 / 0.45 | 0.29 / 0.3 | 0.22 / 0.24 |
| Solar | 0.53 / 0.58 | 0.21 / 0.17 | 0.26 / 0.25 |
| Nuclear | 0.52 / 0.45 | 0.2 / 0.14 | 0.28 / 0.41 |
| Hydro | 0.71 / 0.74 | 0.19 / 0.12 | 0.1 / 0.14 |

**Extended Data Fig. 5 | Comparison of the drivers of primary energy and electricity generation sources between using absolute values or their fractions of the total.** Variance decomposition results in 2050 for the primary energy mix (green shades) and the energy mix used for electricity generation (yellow shades). **Left**: dark (green/yellow) shades indicate the decomposition based on absolute values, while light (green/yellow) shades indicate the decomposition based on fractions of the total. Dashed lines are drawn between absolute/fractional pairs that belong to the same variable. **Right**: tables showing the fractions of the variance explained by each factor (as in Fig. 3 of the main text). Tuples in tables indicate indices by performing the analysis on "absolute values/fractional values". Sorting and blue/red shading of the tables is (still) based on the absolute-value analysis.

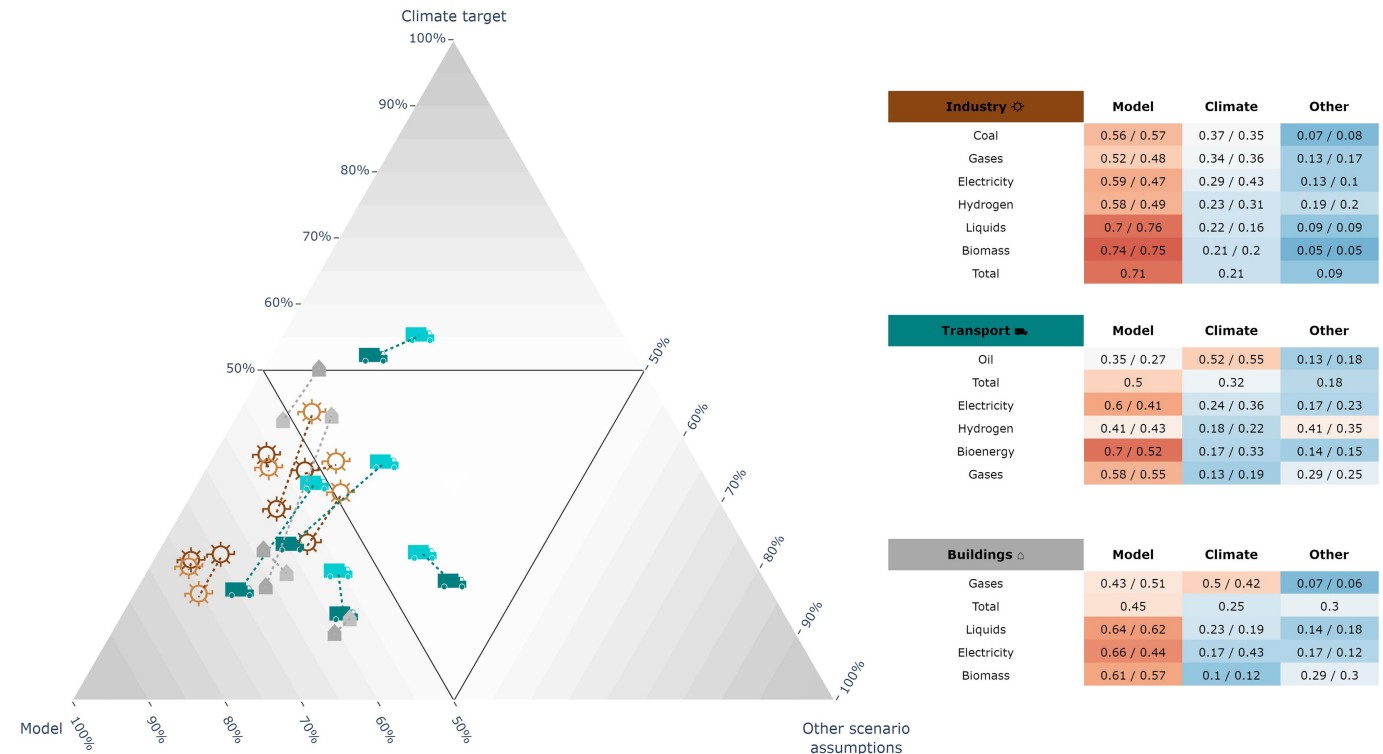

**Extended Data Fig. 6 | Comparison of the drivers of energy sources in end-use sectors between using absolute values or their fractions of the total.** Analogous to Extended Data Fig. 5, but for energy consumption in end-use sectors.

Secondary Energy|Electricity|Renewables (incl. Biomass) (fr) in 2100

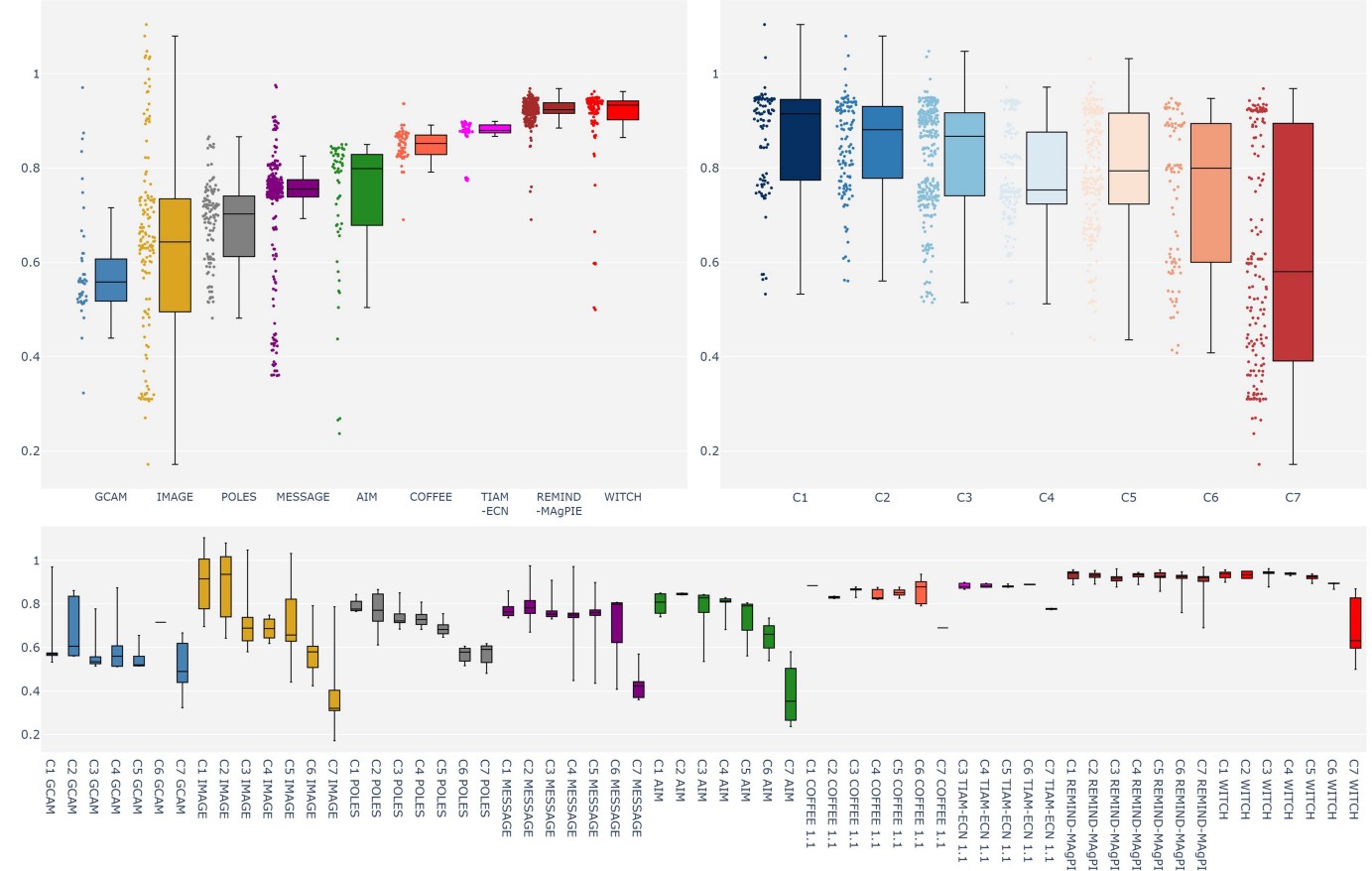

**Extended Data Fig. 7 | Projections of renewables (including) biomass in 2100, as fraction of the total electricity generation, shown per model and climate category.** Analogous to Extended Data Fig. 2, but for renewables (including biomass) used for electricity generation in 2100 in fractions of the total electricity generation.

Final Energy|Transportation|Electricity in 2050

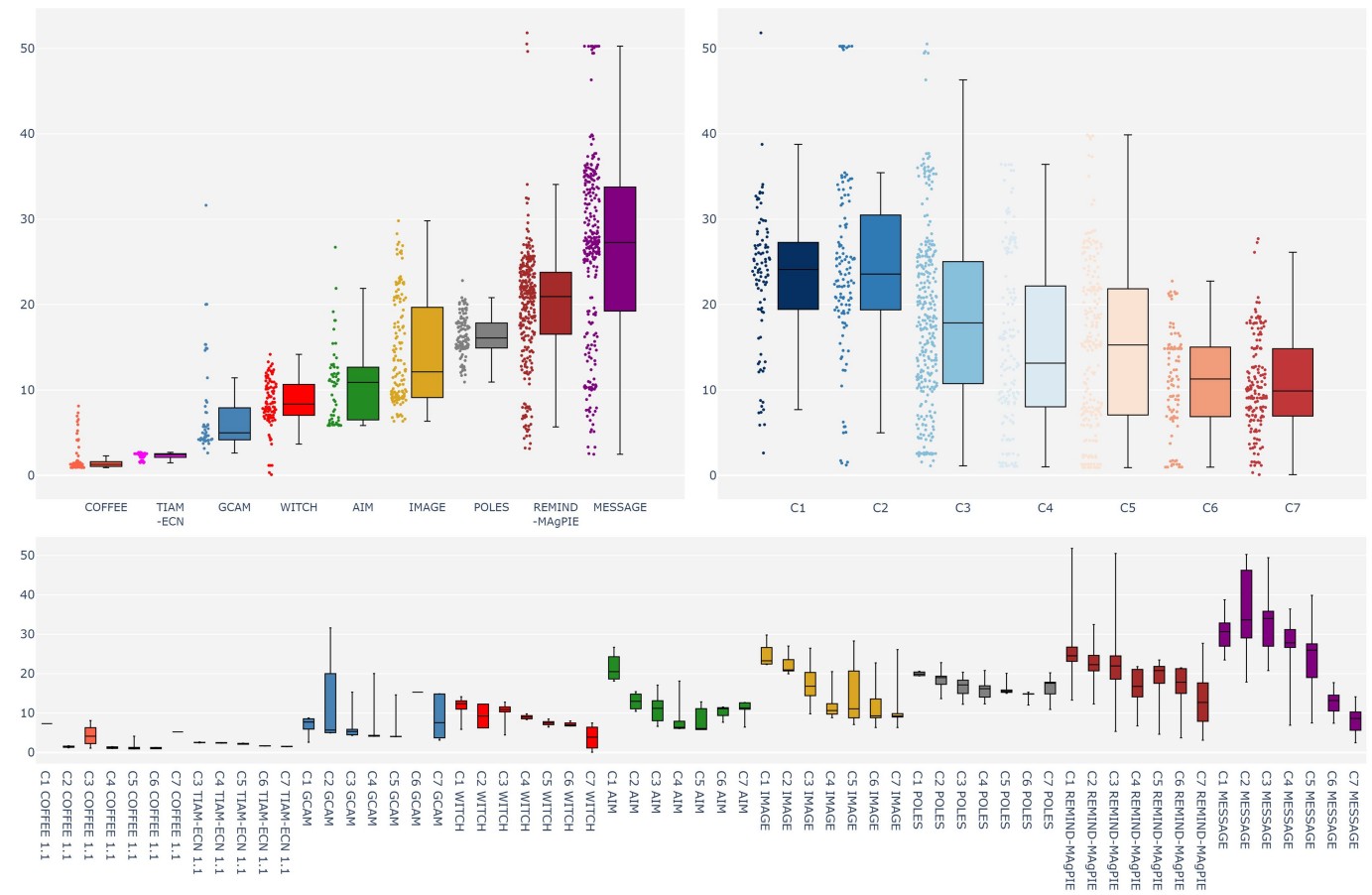

**Extended Data Fig. 8 | Projections of electricity use in the transport sector in 2050, shown per model and climate category.** Analogous to Extended Data Fig. 2, but for electricity use in transport in 2050 (in absolute values).

Final Energy|Transportation|Hydrogen in 2100

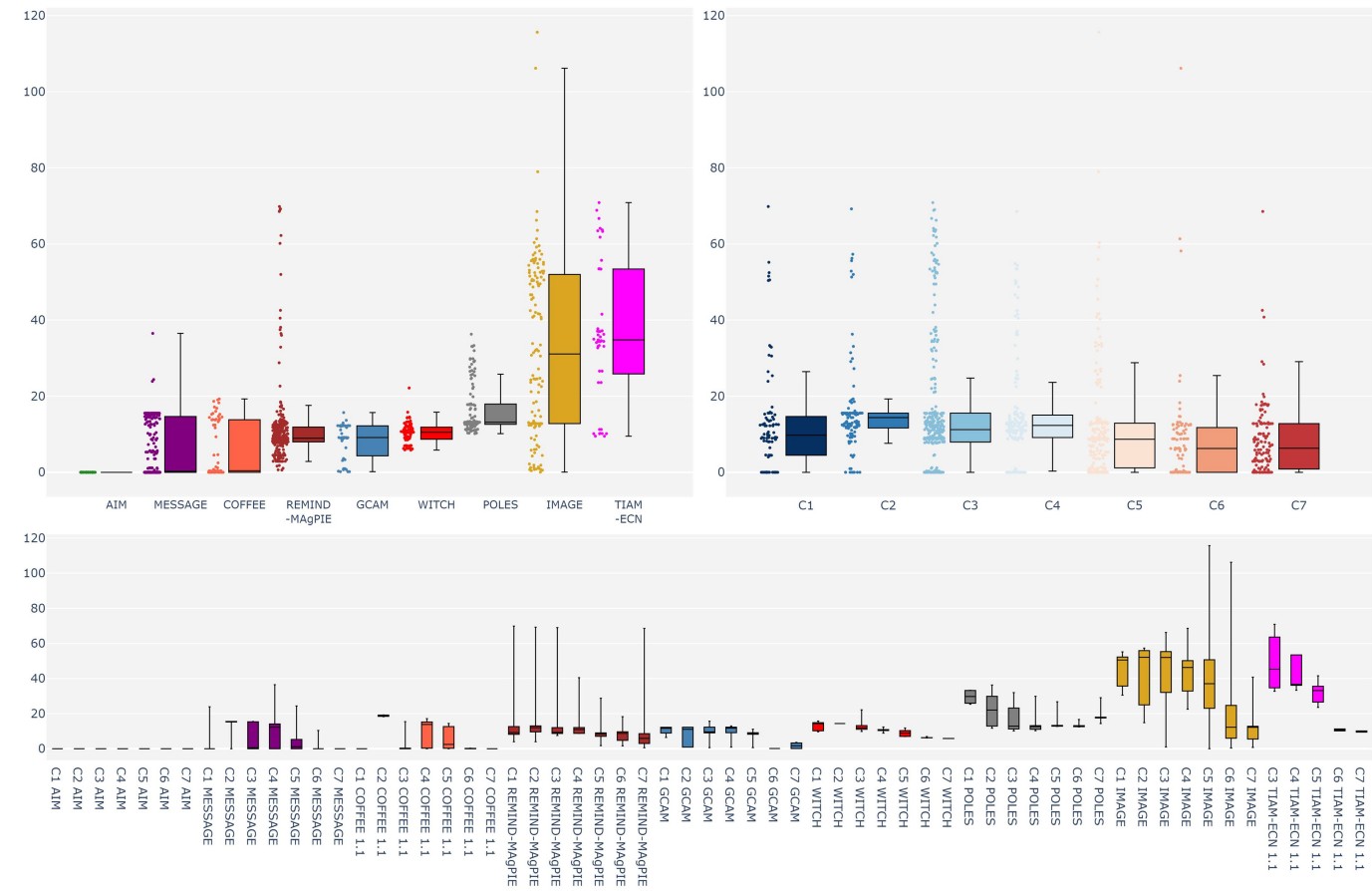

**Extended Data Fig. 9 | Projections of hydrogen use in the transport sector in 2100, shown per model and climate category.** Analogous to Extended Data Fig. 2, but for hydrogen use in transport in 2100 (in absolute values).

Final Energy|Transportation|Liquids|Bioenergy in 2050

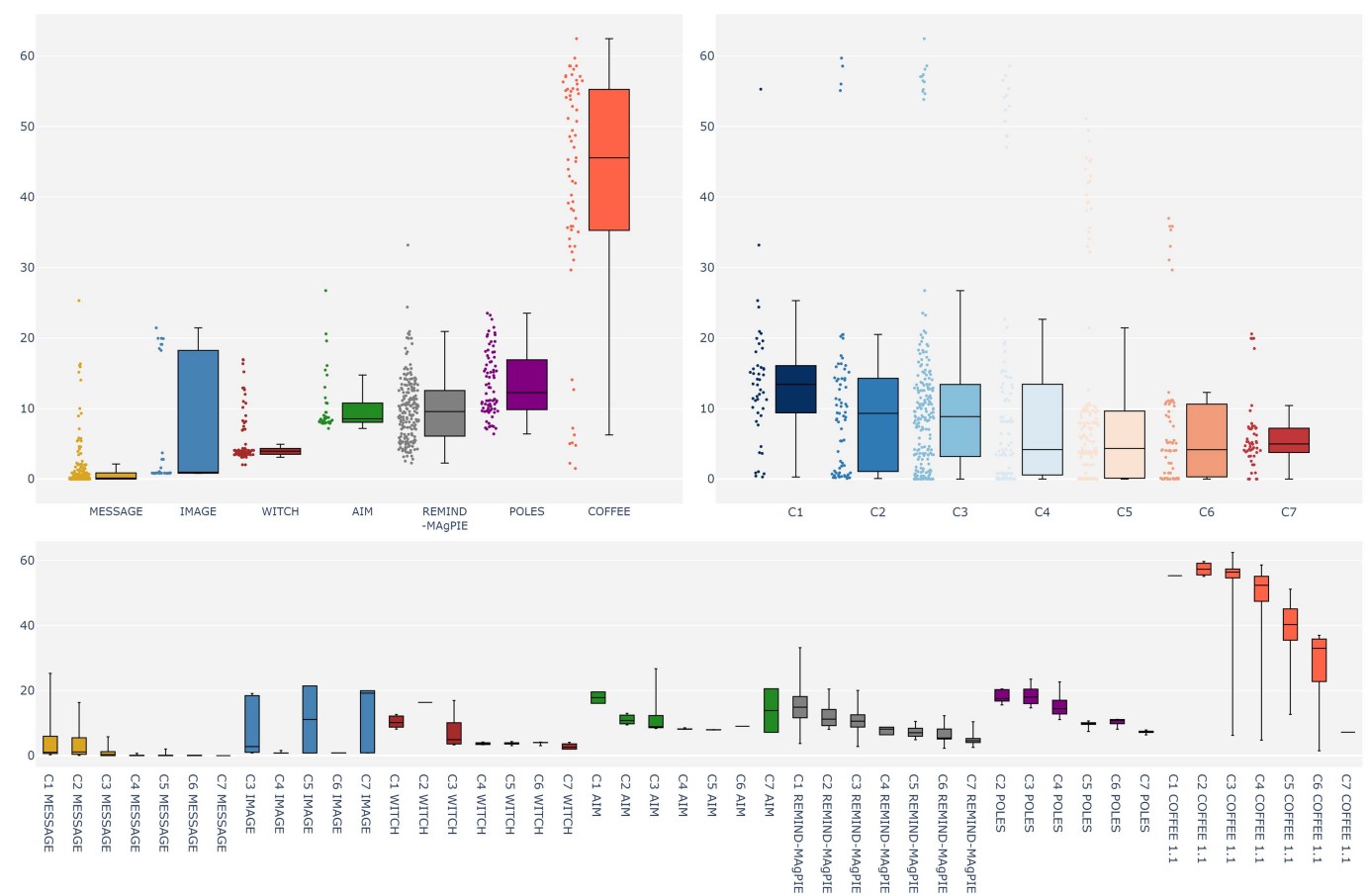

**Extended Data Fig. 10 | Projections of liquid biofuel use in the transport sector in 2050, shown per model and climate category.** Analogous to Extended Data Fig. 2, but for bioenergy use in transport in 2050 (in absolute values).

**Extended Data Table 1 | IPCC climate categories and their definitions**

| CLIMATE CATEGORY | MEANING |
| --- | --- |
| C1 | Limit warming to 1.5 °C (>50%) with no or limited overshoot |
| C2 | Return warming to 1.5 °C (>50%) after a high overshoot |
| C3 | Limit warming to 2 °C (>67%) |
| C4 | Limit warming to 2 °C (>50%) |
| C5 | Limit warming to 2.5 °C (>50%) |
| C6 | Limit warming to 3 °C (>50%) |
| C7 | Limit warming to 4 °C (>50%) |
| C8 | Exceed warming of 4 °C (>=50%) |

**Extended Data Table 2 | Model details**

| ABBR | MODEL | VERSIONS | INSTITUTE |
|------|-------|----------|-----------|
| IMA | IMAGE | IMAGE 3.0<br>IMAGE 3.0.1<br>IMAGE 3.0.2<br>IMAGE 3.2 | Netherlands Environmental Assessment Agency (PBL) |
| TIA | TIAM-ECN | TIAM-ECN 1.1 | Netherlands Organization for applied scientific research (TNO) |
| GCA | GCAM | GCAM 4.2<br>GCAM 5.2<br>GCAM 5.3<br>GCAM-PR 5.3 | Joint Global Change Research Institute (JGCRI) |
| POL | POLES | POLES ADVANCE<br>POLES CD-LINKS<br>POLES EMF30<br>POLES EMF33<br>POLES ENGAGE<br>POLES GECO2019 | Joint Research Council (JRC) |
| AIM | AIM | AIM/CGE 2.0<br>AIM/CGE 2.1<br>AIM/CGE 2.2<br>AIM/Hub-Global 2.0 | National Institute for Environmental Studies (NIES) |
| MES | MESSAGE(ix-GLOBIOM) | MESSAGE-GLOBIOM 1.0<br>MESSAGEix-GLOBIOM 1.0<br>MESSAGEix-GLOBIOM_1.1<br>MESSAGEix-GLOBIOM_1.2<br>MESSAGEix-GLOBIOM_GEI 1.0 | International Institute for Applied Systems Analysis (IIASA) |
| REM | REMIND | REMIND 1.6<br>REMIND 1.7<br>REMIND 2.1<br>REMIND-Buildings 2.0<br>REMIND-MAgPIE 1.5<br>REMIND-MAgPIE 1.7-3.0<br>REMIND-MAgPIE 2.0-4.1<br>REMIND-MAgPIE 2.1-4.2<br>REMIND-MAgPIE 2.1-4.3<br>REMIND-Transport 2.1 | Potsdam Institute for Climate Impact Research (PIK) |
| WIT | WITCH | WITCH 4.6<br>WITCH 5.0<br>WITCH-GLOBIOM 3.1<br>WITCH-GLOBIOM 4.2<br>WITCH-GLOBIOM 4.4 | Euro-Mediterranean Center for Climate Change (CMCC) |
| COF | COFFEE | COFFEE 1.1 | Alberto Luiz Coimbra Institute for Graduate Studies and Research in Engineering (COPPE/UFRJ) |
| CRO | C-ROADS | C-ROADS-5.005 | Climate Interactive |
| EPP | EPPA | EPPA 6 | Massachusetts Institute of Technology (MIT) |
| GEM | GEM-E3 | GEM-E3_V2021 | Institute of Communication and Computer Systems (ICCS) |
| MER | MERGE-ETL | MERGE-ETL 6.0 | Paul Scherrer Institute (PSI) |

The table contains model abbreviations, versions that are merged in this analysis and the associated institute managing the model. Models colored in orange are excluded from this analysis because of their small abundance in the dataset. For more information on each model, the reader is referred to the Integrated Assessment Modelling Community (IAMC) wiki: https://www.iamcdocumentation.eu/index.php/IAMC_wiki.

**Extended Data Table 3 | Model and climate category prevalence**

|  | C1 | C2 | C3 | C4 | C5 | C6 | C7 | C8 | Total |
|---|---|---|---|---|---|---|---|---|---|
| *AIM* | 4 | 3 | 17 | 8 | 13 | 4 | 6 | 0 | 55 |
| *COFFEE* | 1 | 4 | 14 | 15 | 21 | 9 | 1 | 0 | 65 |
| *EPPA* | 0 | 0 | 1 | 3 | 0 | 1 | 2 | 0 | 7 |
| *GCAM* | 6 | 5 | 13 | 9 | 6 | 1 | 6 | 1 | 47 |
| *IMAGE* | 7 | 9 | 34 | 18 | 22 | 16 | 34 | 2 | 142 |
| *MERGE-ETL* | 0 | 0 | 0 | 1 | 0 | 0 | 0 | 0 | 1 |
| *MESSAGE(ix-GLOBIOM)* | 20 | 43 | 59 | 39 | 57 | 20 | 28 | 0 | 266 |
| *POLES* | 4 | 10 | 26 | 24 | 20 | 11 | 19 | 0 | 114 |
| *REMIND(-MAgPIE)* | 41 | 44 | 84 | 16 | 34 | 19 | 48 | 11 | 297 |
| *TIAM-ECN* | 0 | 0 | 20 | 6 | 10 | 4 | 5 | 0 | 45 |
| *WITCH* | 9 | 2 | 29 | 14 | 24 | 9 | 12 | 14 | 113 |
| *Total* | 92 | 120 | 297 | 153 | 207 | 94 | 161 | 28 | 1152 |

Orange rows and columns are removed from the analysis because of limited prevalence. Blue cells indicate imperfections of the ensemble that we accepted in the analysis.

**Extended Data Table 4 | Higher order terms for the variables used in Fig. 1, concerning electricity generation**

| Category | Variable | Model-climate term ($S_{cm}$) (second-order) | | | Other scenario assumptions ($F_o$) (multi-order) | | |
|---|---|---|---|---|---|---|---|
| | | 2030 | 2050 | 2100 | 2030 | 2050 | 2100 |
| **Fossil use** | Total | 0.09 | 0.14 | 0.25 | 0.24 | 0.11 | 0.14 |
| | Coal (without CCS) | 0.08 | 0.16 | 0.26 | 0.24 | 0.12 | 0.19 |
| | Gas (without CCS) | 0.12 | 0.15 | 0.35 | 0.30 | 0.18 | 0.16 |
| | | | | | | | |
| **Renewables** | Total | 0.44 | 0.15 | 0.11 | 0.10 | 0.18 | 0.23 |
| | Solar | 0.39 | 0.13 | 0.13 | 0.13 | 0.26 | 0.25 |
| | Wind | 0.46 | 0.19 | 0.06 | 0.10 | 0.16 | 0.2 |
| | Hydro | 0.24 | 0.16 | 0.14 | 0.14 | 0.10 | 0.11 |
| | Biomass (without CCS) | 0.16 | 0.19 | 0.18 | 0.20 | 0.16 | 0.28 |
| | | | | | | | |
| **CCS** | Total | 0.12 | 0.12 | 0.17 | 0.39 | 0.41 | 0.20 |
| | Coal (with CCS) | 0.26 | 0.18 | 0.21 | 0.25 | 0.21 | 0.40 |
| | Gas (with CCS) | 0.23 | 0.22 | 0.22 | 0.27 | 0.22 | 0.18 |
| | Biomass (with CCS) | 0.28 | 0.25 | 0.23 | 0.29 | 0.26 | 0.16 |

The first column ("Category") is refers to the three categories (red, blue and green colors) used in Fig. 1. A few values discussed in the text below are highlighted in blue.