## [Peer Review File · Nature]

Manuscript Title: Spread in climate policy scenarios unraveled

Reviewer Comments & Author Rebuttals

Reviewer Reports on the Initial Version:

Referees' comments:

Referee #1 (Remarks to the Author):

The paper presents an analysis of the AR6 scenario ensemble, exploring whether specific scenario characteristics and mitigation responses are dependent on the climate target or rather on the underlying model or scenario assumptions. For this purpose the authors apply a sobol decomposition to the AR6 scenario ensemble. The main conclusion is that many scenario characteristics depend rather on the model than the climate target.

The paper presents in my view very important insight with high policy relevance. I have though some concerns that I think critically influence the results and which I would like the authors to clarify:

- The "Scenario" dimension": Exploring the dependency of the results in terms of the scenario dimension seems to me tautological, since the vast majority of the scenarios in AR6 are based on the same scenario narrative (SSP2, depicting middle of the road scenario assumptions). This in turn means that the AR6 scenarios are designed around similar assumptions (in qualitative and quantitative terms). The fact is well known and described in the underlying scenario chapter of the AR6 WGIII (chapter 3). Decomposing this dimension thus does not yield any new insights. I would argue that it actually increases the complexity of the analysis and figures unnecessarily.

The more interesting conclusions are derived from the fact that many of the results depend on the models rather than the climate target. These results should be decomposed in my view more cleanly rather than adding the third dimension that is not represented well in the set (and thus also not clear what the Sobol decomposition actually means if the input parameters are not representative).

- Model biases: I am wondering whether the over-representation of individual models in the AR6 is problematic. Roughly half of the scenarios in the AR6 seem to come from two models only. This over-representation will have an effect on the conclusion that many of the mitigation outcomes are model dependent (rather than a function of the climate target). It seems to me necessary thus to debias the sample first, which is common practice for example in the physical climate sciences. I.e., the bias-corrected sample should comprise an equal representation of scenarios by each model before doing the sobol decomposition. Otherwise, I think the conclusions are biased and not generalizable in a wider context and are only valid for the properties of the AR6 sample (basically also meaning that the whole conclusions about the limited robustness of the insights on wind, hydro and other technologies would not hold).

- It is well know that some of the scenario results, particularly for individual options, are less robust

and thus more dependent on the model. In order to better understand the results it would be important to give some indication why specific models have such a big influence on specific outcomes. This will differ for different scenario characteristics. But is the model dependency due to the different modelling paradigms (eg, optimization vs simulation), differences in model structure, model constraints/ parameters, etc...?

The paper is overall interesting and important. I hope thus that the authors will be able to revise and resubmit the paper.

Referee #2 (Remarks to the Author):

SUMMARY AND INTRO

The manuscript by Dekker et al. contributes to the literature on climate change scenarios. As the authors note, climate change scenarios are critical inputs to discussions around climate change futures and potential policy actions, most recently highlighted by the inclusion of a wide set of such scenarios in the latest IPCC report as stored in the CMIP6 database (climate projections) and AR6 database (climate scenarios). These often highly detailed scenarios cover a wide range of possible futures, motivating the authors to take on a detailed Sobol method^{1,2} of global sensitivity analysis to decompose the variance of key variables into contributions by three defined sources: climate targets, model differences, and scenario assumptions. The methodology departs slightly from the standard presentation of Sobol indices (a full decomposition into first, second, and total-order indices) due to modeling limitations as described in the Methods section.

The manuscript divides results into a few primary sections: electricity generation variables, transport energy demand variables, and an overview of variables in 2050. They find that the variance of most variables is attributable to either model differences or climate targets, and less so to scenario assumptions. They also discuss the importance and policy implications of a “consensus” result (defined as “aspects of the energy transition that are robustly linked to climate targets, and less driven by model differences and scenario assumptions”).

My overall read of this paper is that it is an important application of state-of-the-art global sensitivity analysis methods to a set of modeled climate scenarios. These scenarios do play a central role in both academic research and public policy. I am however very concerned that the level of domain detail and assumptions of reader knowledge combined with a lack of clarity about the central findings and their relative importance, might preclude this work from being appropriate for Nature. I believe it may be a better fit for a more domain-specific specialist audience, but would defer to the

Editor and consensus with other Reviewers. Below I will detail some major and minor suggested revisions.

INTUITION AND KEY IMPLICATIONS

The introduction of the paper does not provide the readers with enough guidance on intuition to comprehend the significance of the next few sections. The second and third paragraphs contain some language which do so, but the intuition behind how to interpret the attribution and consensus is too vague and scattered throughout the paper. I understand that some of this should be saved for the conclusions, but believe it needs to be better consolidated into the introduction so as to guide the reading. Another option might be to begin each of the three main content sections with a highlight of the primary finding, and related implication, of that section, before diving into the details.

The conclusions summarize the main points generally quite well, including implications of (1) the variables found to be robustly linked to climate targets (2) variables whose variance is primarily due to model choice and (3) the importance of the lack of variable variance dominated by, or even partially influenced by, other scenario assumptions. That said, I think there is a bit more clarity needed on what the authors see as implications for the modeling community. For example, variables with consensus show that across models and assumptions, certain conclusions about variables are robust and primarily attributable to climate target. That said, the authors also point out that the very little variance being attributed to scenario assumptions shows that perhaps the set of assumptions represented is too narrow. Put another way, is there a result that the authors would like to see in the future, or are they choosing not to comment on that for the modeling community and instead primarily advise on the use cases and implications for users?

CLARITY, DEFINITIONS, AND EXAMPLES

For this paper to fit the broad audience of Nature the authors should add some clarity to their definitions and vernacular. This is especially necessary in the introduction so as not to confuse or lose readers. First, it would be useful to have a well-defined definition of a “climate scenario”. This can of course be short, but a general reader might not be clear on what that does (or does not) encompass so a definition and perhaps toy example/diagram would be helpful to set up the paper. Relating this description to the three categories of variance attribution (climate targets, model differences, and scenario assumptions) would help to frame the paper. The reader comes away with a lot of terminology but may have trouble linking these pieces together. Lastly, the authors consistently use the terms “consensus” and “robust(ness)”, the first of which is formally defined in the introduction but the second of which seems to be used frequently, perhaps as a synonym in this context. In light of literature around robust decision making, or robustness in a statistical setting, it would be helpful to be clear on what the authors mean by a “robust” result. The terms robust and consensus have a general and positive tone, so some explanation for why the authors have chosen to focus these words on climate target would be useful. I note the focus on this dimension for the

trio makes sense to me, so I do not disagree, simply think it would help a reader to have a line or two to clarify it. These of course are my opinions, and perhaps to others these introductions are clearer.

FIGURES

I find Figures 2 and 3 to include too much information, along too many dimensions, to tell a clear story. I do appreciate the significant effort to include these dimensions and the creativity used to do so, but worry they are simply too overwhelming. I would encourage the authors to consider either splitting these figures out into panels to subset one of the variables, or removing all together a dimension or two if it is not a dominant theme in the conclusions. For example, I personally find the use of width for as coefficient of variance, and tracing of the variables over time using a continuous line, to be fairly uninformative since they are hard to interpret by eye. I would be quite flexible on solutions to this concern, so leave that to the authors.

METHODS

The Sobol analysis method is often touted for its ability to decompose uncertainty not only to first order effects, but also give key insights to second order (and other higher order) indices, summing finally to total order indices. While I understand the limitations that lead to needing to adapt this framework described in the methodology, I was surprised that there was no attempt to dissect second order effects, for example the interaction relationship between model differences and climate targets. Did the authors look into this and find it uninformative? Or perhaps was it too complex to try to include in the overall messaging of this paper? Some explanation would be useful.

I would request more information on the choice of the number of trials and number of resamples, more than the one comparison to sample size of 1000 and resample size of 30. This could come in the form of a convergence graph showing how indices change as these parameters change, or explicit confidence intervals, or something else. I think this would help give confidence in the stability and convergence of the results.

REFERENCES

1 Saltelli, A. Making best use of model evaluations to compute sensitivity indices. *Computer Physics Communications* 145, 280-297 (2002). [https://doi.org:https://doi.org/10.1016/S0010-4655\(02\)00280-1](https://doi.org/https://doi.org/10.1016/S0010-4655(02)00280-1)

2 Sobol, I. M. Sensitivity Estimates for Nonlinear Mathematical Models. *Mathematical Modelling and Computational Experiments* 4, 407-414 (1993).

Referee #3 (Remarks to the Author):

This article introduces a novel methodology to evaluate different sources of uncertainty and variance across climate policy scenarios. This is a very important project as it begins to quantify the contribution of different factors to scenario uncertainty across different sectors - whether it be targets, model parameters, or scenario assumptions among others. This is helpful to the scientific community in pointing out areas of stronger vs weaker scientific consensus and outlining a research agenda for where uncertainty/variance needs to be more constrained. Secondly, it provides policymakers with a stronger foundation for decision-making in certain sub-fields of climate policy. In general, I recommend it for publication if they can address one significant issue described below.

My main comment is with regards to the figures, especially 1 and 2. They are extremely hard to understand and interpret in their current form. Perhaps this is due to the novel (for this field) methodology being used, but the lines in the triangle are very hard to decipher - their thickness and direction are not well explained, only one year is listed, and their overlap makes it very difficult to tease apart the different factors they're discussing. Perhaps this wouldn't be as big of an issue if this article was submitted to a more discipline- or methodology-specific journal, but the fact that this article is meant for a more general scientific audience in Nature means that the figure needs to be made clearer. This could mean either splitting out the figure into separate figures or using a different format for communicating the results. The central messages of the figures need to be evident relatively easily and quickly for the reader.

Author Rebuttals to Initial Comments:

Referee #1 (Remarks to the Author):

The paper presents an analysis of the AR6 scenario ensemble, exploring whether specific scenario characteristics and mitigation responses are dependent on the climate target or rather on the underlying model or scenario assumptions. For this purpose the authors apply a Sobol decomposition to the AR6 scenario ensemble. The main conclusion is that many scenario characteristics depend rather on the model than the climate target.

The paper presents in my view very important insight with high policy relevance. I have though some concerns that I think critically influence the results and which I would like the authors to clarify:

[#1.1] The "Scenario dimension": Exploring the dependency of the results in terms of the scenario dimension seems to me tautological, since the vast majority of the scenarios in AR6 are based on the same scenario narrative (SSP2, depicting middle of the road scenario assumptions). This in turn means that the AR6 scenarios are designed around similar assumptions (in qualitative and quantitative terms). The fact is well known and described in the underlying scenario chapter of the AR6 WGIII (chapter 3). Decomposing this dimension thus does not yield any new insights. I would argue that it actually increases the complexity of the analysis and figures unnecessarily.

The more interesting conclusions are derived from the fact that many of the results depend on the models rather than the climate target. These results should be decomposed in my view more cleanly rather than adding the third dimension that is not represented well in the set (and thus also not clear what the Sobol decomposition actually means if the input parameters are not representative).

We believe scenario differences are an additional, critical factor that could explain variance – and, therefore, needs to be included. Formally, we qualify this by attributing the unexplained variance, given the allocation to scenario target and model. While part of this variance is driven by the baseline scenario uncertainty (SSP2 versus other scenarios) as pointed out by the reviewer, other scenario variations including different timing of mitigation action, NDCs, specific policies, and different SPAs could have played a role here as well. The finding is that the influence of this factor (beyond model and scenario target) is relatively small.

Even when realizing that part of the causes of low variation in the “other scenario assumptions” dimension is due to the dominance of one scenario narrative (SSP2) in the database, we believe this dimension is important to keep in the results. First, the low scenario variation might not be known for all users, but above the impact of this bias has not been quantified before. In that context, the results nicely illustrate that further scenario diversity is a critical factor for future work, which we mentioned this already in the manuscript (see cond-to-last paragraph).

Second, it is important to stress that for many variables, the impact of scenario assumptions is not negligible, nor uniform across variables. For example, more than 40% of the variance of hydrogen use projections in transport (in 2050) is explained by other factors than model and climate target which is high compared to other variables and valuable information that sheds light on the relative importance of the climate target and model choice. Also, we have added the total energy use for electricity generation that involve CCS to the new figure 1 (in blue), which also shows significant values in this dimension. We will adjust the color in the tables in Fig. 3 in the next version to show the non-uniformity in this dimension better.

Finally, there is also a contribution to higher order terms as noted in the Methods – on which the referee rightfully notes that it may complicate the interpretation of this term. To explain this better for a more technical reader, we now dedicate two new supplementary sections (SI A.5 and A.6) to

discussing higher order terms in general and provide illustrative examples of the “other scenario assumptions” dimension.

By now making clearer what the added value of the third dimension is, and by simplifying the Figures (see also in other responses to referee comments) we consider ourselves on the right side of the balance involved in the complexity, but also the comprehensiveness of adding this dimension.

Changes:

- We added new supplementary sections (SI A.5-A.6) that discuss higher order terms and the composition of the “Other scenario assumptions” dimension.
- We changed the coloring of the table columns in Fig. 3 to clarify contrast between high and low numbers (notably for the “Other scenario assumptions” dimension). Blue is now below 33%, and red is above 33%, being a natural division of the three factors.
- We elaborate more on the “Other scenario assumptions” dimension in the beginning and final part of the main text and refer to the new SI A.6 section.
- We significantly modified the triangular figures to be more easy to understand, also including a total CCS use in electricity generation that is significantly affected by other scenario assumptions.

[#1.2] Model biases: I am wondering whether the over-representation of individual models in the AR6 is problematic. Roughly half of the scenarios in the AR6 seem to come from two models only. This over-representation will have an effect on the conclusion that many of the mitigation outcomes are model dependent (rather than a function of the climate target). It seems to me necessary thus to debias the sample first, which is common practice for example in the physical climate sciences. I.e., the bias-corrected sample should comprise an equal representation of scenarios by each model before doing the Sobol decomposition. Otherwise, I think the conclusions are biased and not generalizable in a wider context and are only valid for the properties of the AR6 sample (basically also meaning that the whole conclusions about the limited robustness of the insights on wind, hydro and other technologies would not hold).

The referee addresses an important point. In fact, we already debiased the sample exactly like the referee suggested – see “Variance decomposition” in the Methods section. After a number of pre-processing steps (see “Pre-processing”), we draw samples of the dataset in which models and climate targets are equally represented. For each model-climate target pair, we draw 3000 members, naturally involving replacing. This way, e.g., the REMIND model has exactly as many entries as the IMAGE model does, and the C1 climate target has equally many entries as the C3 target. We redraw and calculate the results 100 times, and then report the average, to limit the role of chance in determining the final conclusions. We now emphasized this more in the main text. Note that we also included a sensitivity analysis with respect to the parameters of this sampling procedure in SI A.4.

Changes:

- In the main text, we added a reference to the debiasing procedure.

[#1.3] It is well known that some of the scenario results, particularly for individual options, are less robust and thus more dependent on the model. In order to better understand the results it would be important to give some indication why specific models have such a big influence on specific outcomes. This will differ for different scenario characteristics. But is the model dependency due to the different modelling paradigms (e.g., optimization vs simulation), differences in model structure, model constraints/ parameters, etc.?

The causes of impact of model differences has been studied for specific model aspects in the past. For example, Daioglou et al. (2020) investigated whether parametrization or model solution paradigm influenced long-term bioenergy projections¹ and Koelbl et al. (2014) researched the causes of model differences in Carbon Capture and Storage projections². We will add these references to the main text. These studies do emphasize that so many factors play a role at the same time – they have highlighted that model outcomes do not seem to depend on techno-economic parameterization, type of model, or solution methods. In other words, it is really difficult to identify specific factors. Instead, differences in model outcomes may be to more heterogeneous aspects that are difficult to capture in labels: e.g., how energy and land systems are represented, their technology aggregation, projections of demand for energy/agriculture and the elasticity of this demand, and how system inertia is represented. These papers highlight that such investigations and also their conclusions are highly specific to each variable. Hence, analyzing this for *all* variables (and associated dynamics) would be beyond the scope of this paper (yielding individual papers on their own).

Realizing the complexity of causes of model differences and how this differs per variable, and balancing this with the scope of this paper, we have decided to address this topic by referring to a number of references that already research this question on specific variables^{1,2}, as well as overall model diagnostics papers³⁻⁵.

Changes:

- We address the discussion on causes of model differences now in the main text and included references of previous studies on this matter.

The paper is overall interesting and important. I hope thus that the authors will be able to revise and resubmit the paper.

We thank the referee for his/her useful comments.

Referee #2 (Remarks to the Author):

SUMMARY AND INTRO

The manuscript by Dekker et al. contributes to the literature on climate change scenarios. As the authors note, climate change scenarios are critical inputs to discussions around climate change futures and potential policy actions, most recently highlighted by the inclusion of a wide set of such scenarios in the latest IPCC report as stored in the CMIP6 database (climate projections) and AR6 database (climate scenarios). These often highly detailed scenarios cover a wide range of possible futures, motivating the authors to take on a detailed Sobol method^{6,7} of global sensitivity analysis to decompose the variance of key variables into contributions by three defined sources: climate targets, model differences, and scenario assumptions. The methodology departs slightly from the standard presentation of Sobol indices (a full decomposition into first, second, and total-order indices) due to modeling limitations as described in the Methods section.

The manuscript divides results into a few primary sections: electricity generation variables, transport energy demand variables, and an overview of variables in 2050. They find that the variance of most variables is attributable to either model differences or climate targets, and less so to scenario assumptions. They also discuss the importance and policy implications of a “consensus” result (defined as aspects of the “energy transition that are robustly linked to climate targets, and less driven by model differences and scenario assumptions”).

My overall read of this paper is that it is an important application of state-of-the-art global sensitivity analysis methods to a set of modeled climate scenarios. These scenarios do play a central role in both academic research and public policy. I am however very concerned that the level of domain detail and assumptions of reader knowledge combined with a lack of clarity about the central findings and their relative importance, might preclude this work from being appropriate for Nature. I believe it may be a better fit for a more domain-specific specialist audience, but would defer to the Editor and consensus with other Reviewers. Below I will detail some major and minor suggested revisions.

We thank the referee for his/her useful comments.

INTUITION AND KEY IMPLICATIONS

[#2.1] The introduction of the paper does not provide the readers with enough guidance on intuition to comprehend the significance of the next few sections. The second and third paragraphs contain some language which do so, but the intuition behind how to interpret the attribution and consensus is too vague and scattered throughout the paper. I understand that some of this should be saved for the conclusions, but believe it needs to be better consolidated into the introduction so as to guide the reading. Another option might be to begin each of the three main content sections with a highlight of the primary finding, and related implication, of that section, before diving into the details. The conclusions summarize the main points generally quite well, including implications of (1) the variables found to be robustly linked to climate targets (2) variables whose variance is primarily due to model choice and (3) the importance of the lack of variable variance dominated by, or even partially influenced by, other scenario assumptions.

Even though the mathematics of the methodology is not of interest to all readers, we think that the results have implications that are of interest to a broad audience. Indeed, as the referee noted, these implications may be difficult to distill from the rather technical writing in the content sections. Therefore, we gratefully adopt two suggestions of the referee. First, we rewrite the introduction to

not only introduce the topic, but also understand what general insights can be obtained from looking at variance – which is not trivial for a broad audience. Second, we adopt the idea of starting each content section with a highlight of the primary finding and a related implication, which we discuss in detail further in that section.

Changes:

- We have rewritten the introduction such that it better introduces the analysis of variance (decomposition) and why this is useful to a broad audience when it comes to climate policy scenarios.
- In the beginning of the sections on Fig 1 and Fig 2, we state the highlights and primary findings up front to guide the reader better.

[#2.2] That said, I think there is a bit more clarity needed on what the authors see as implications for the modeling community. For example, variables with consensus show that across models and assumptions, certain conclusions about variables are robust and primarily attributable to climate target. That said, the authors also point out that the very little variance being attributed to scenario assumptions shows that perhaps the set of assumptions represented is too narrow. Put another way, is there a result that the authors would like to see in the future, or are they choosing not to comment on that for the modeling community and instead primarily advise on the use cases and implications for users?

We believe the merits of this paper are threefold:

- (1) It provides a quantitative overview of the drivers of variance ('consensus in climate policy scenarios'), which clearly shows the model dependency of many crucial variables such as individual renewable and CCS sources.
- (2) It points out the lack of variation stemming from scenario differentiation.
- (3) It contains a (novel) application of this (existing) methodology to this field and argues that this could be a welcome addition to future MMC analyses in mitigation literature.

While mostly (1) and (2) are of high importance to be aware of for a broad audience (including policymakers), all three have implications specific for the modeling community. Merit (1) reveals new areas of research – e.g., identifying areas in which models are distinctly different. For modelers, merit (2) emphasizes the importance of using SSPs beyond only SSP2 as scenario bases and to put more effort in designing alternative scenario narratives in order to really explore the full mitigation solution space. Merit (3) directly addresses the modelers by proposing this framework as a new model intercomparison tool.

We agree with the referee that currently, the paper mostly *states* these merits (for both a broad audience as well as for modelers specifically), without providing specific advice or examples on the implications. We now changed that.

Changes:

- We spelled out the three merits and their implications for both a broad audience, as well as for the modeling community.
- Specifically, we complemented the discussions (also those added in #2.1) with advice to the modelling community.

CLARITY, DEFINITIONS, AND EXAMPLES

[#2.3] For this paper to fit the broad audience of Nature the authors should add some clarity to their definitions and vernacular. This is especially necessary in the introduction so as not to confuse or lose readers. First, it would be useful to have a well-defined definition of a “climate scenario”. This can of course be short, but a general reader might not be clear on what that does (or does not) encompass so a definition and perhaps toy example/diagram would be helpful to set up the paper.

Changes:

- Done. We added a short description of “climate scenario” to the introduction.

[#2.4] Relating this description to the three categories of variance attribution (climate targets, model differences, and scenario assumptions) would help to frame the paper. The reader comes away with a lot of terminology but may have trouble linking these pieces together.

Changes:

- Done. The three categories of variance attribution are now better introduced in both the introduction as well as in the section about Fig. 1.

[#2.5] Lastly, the authors consistently use the terms “consensus” and “robust(ness)”, the first of which is formally defined in the introduction but the second of which seems to be used frequently, perhaps as a synonym in this context. In light of literature around robust decision making, or robustness in a statistical setting, it would be helpful to be clear on what the authors mean by a “robust” result. The terms robust and consensus have a general and positive tone, so some explanation for why the authors have chosen to focus these words on climate target would be useful. I note the focus on this dimension for the trio makes sense to me, so I do not disagree, simply think it would help a reader to have a line or two to clarify it. These of course are my opinions, and perhaps to others these introductions are clearer.

We agree on the suggestion to make this more clear and consistent. Currently, we use these two words synonymously. There is no (intentional) value behind our use of these words. As the referee notes in his/her summary (above), we use these words for variables when they are mostly determined by the climate target rather than the model used or other scenario assumptions. “Consensus” is already explicitly defined (end of introduction), but “robust(ness)” is not.

Changes:

- We clarified the use of “consensus” and “robust(ness)” in the introduction and changed our use of these words in a number of instances.

FIGURES

[#2.6] I find Figures 2 and 3 to include too much information, along too many dimensions, to tell a clear story. I do appreciate the significant effort to include these dimensions and the creativity used to do so, but worry they are simply too overwhelming. I would encourage the authors to consider either splitting these figures out into panels to subset one of the variables, or removing all together a dimension or two if it is not a dominant theme in the conclusions. For example, I personally find the use of width for as coefficient of variance, and tracing of the variables over time using a continuous line, to be fairly uninformative since they are hard to interpret by eye. I would be quite flexible on solutions to this concern, so leave that to the authors.

We agree with the referee on this point, which is also noted by other referees. We have come up with a list of changes to the figures that we think will facilitate more easy reading and understanding of the

figures, as well as appealing to a broader audience. Part of this list is indeed the removal of showing the coefficient of variance with the line width and making individual years more clearly visible rather than using continuous lines.

Changes:

- We significantly modified all three figures in the main text as well as a number of figures in the supplementary material. In particular, we removed the use of line width as a measure of the coefficient of variance, we reduced all temporal data to only 2030, 2050 and 2100, and made visual improvement such as reducing the amount of colors (guiding the reader more towards the main messages).

METHODS

[#2.7] The Sobol analysis method is often touted for its ability to decompose uncertainty not only to first order effects, but also give key insights to second order (and other higher order) indices, summing finally to total order indices. While I understand the limitations that lead to needing to adapt this framework described in the methodology, I was surprised that there was no attempt to dissect second order effects, for example the interaction relationship between model differences and climate targets. Did the authors look into this and find it uninformative? Or perhaps was it too complex to try to include in the overall messaging of this paper? Some explanation would be useful.

Without showing the results, we did in fact look into this. We did not report it, because it made the paper more complex for relatively little additional insight in relation to the general scope of the paper. The most important observation we got from that analysis is that the second-order interaction between model differences and climate targets for several variables was significant (up to 30% even; this is mentioned in the Methods section) – which implied that we could not neglect them (which is done in previous work⁸). We agree with the referee that for a certain fraction of readers, insights in higher order terms may be of interest. Therefore, we have now added a section on this in the supplementary material.

Changes:

- We added two new supplementary sections (SI A.5-A.6) in which we discuss the higher order terms, notably the second-order interaction term between model differences and climate targets.

[#2.8] I would request more information on the choice of the number of trials and number of resamples, more than the one comparison to sample size of 1000 and resample size of 30. This could come in the form of a convergence graph showing how indices change as these parameters change, or explicit confidence intervals, or something else. I think this would help give confidence in the stability and convergence of the results.

Changes:

- Done. We added this to a new supplementary section (SI A.4) on the sensitivity of the results with respect to the parameter settings, including the requested convergence graph.

Referee #3 (Remarks to the Author):

This article introduces a novel methodology to evaluate different sources of uncertainty and variance across climate policy scenarios. This is a very important project as it begins to quantify the contribution of different factors to scenario uncertainty across different sectors - whether it be targets, model parameters, or scenario assumptions among others. This is helpful to the scientific community in pointing out areas of stronger vs weaker scientific consensus and outlining a research agenda for where uncertainty/variance needs to be more constrained. Secondly, it provides policymakers with a stronger foundation for decision-making in certain sub-fields of climate policy. In general, I recommend it for publication if they can address one significant issue described below.

We thank the referee for his/her useful comments.

[#3.1] My main comment is with regards to the figures, especially 1 and 2. They are extremely hard to understand and interpret in their current form. Perhaps this is due to the novel (for this field) methodology being used, but the lines in the triangle are very hard to decipher - their thickness and direction are not well explained, only one year is listed, and their overlap makes it very difficult to tease apart the different factors they're discussing. Perhaps this wouldn't be as big of an issue if this article was submitted to a more discipline- or methodology-specific journal, but the fact that this article is meant for a more general scientific audience in Nature means that the figure needs to be made clearer. This could mean either splitting out the figure into separate figures or using a different format for communicating the results. The central messages of the figures need to be evident relatively easily and quickly for the reader.

We agree with the referee and this remark is in line with the other referees. We have made significant changes to the figures (also in the supplementary information) in order to make the more readable for a wide audience, easy to understand and to convey the most important messages better.

Changes:

- We significantly modified all three figures in the main text as well as a number of figures in the supplementary material. In particular, we removed the use of line width as a measure of the coefficient of variance, we reduced all temporal data to only 2030, 2050 and 2100, and made visual improvement such as reducing the amount of colors (guiding the reader more towards the main messages).

References

- 1 Daioglou, V. *et al.* Bioenergy technologies in long-run climate change mitigation: results from the EMF-33 study. *Climatic Change* **163**, 1603-1620 (2020). <https://doi.org/10.1007/s10584-020-02799-y>
- 2 Koelbl, B. S., van den Broek, M. A., Faaij, A. P. C. & van Vuuren, D. P. Uncertainty in Carbon Capture and Storage (CCS) deployment projections: a cross-model comparison exercise. *Climatic Change* **123**, 461-476 (2014). <https://doi.org/10.1007/s10584-013-1050-7>
- 3 Kriegler, E. *et al.* Diagnostic indicators for integrated assessment models of climate policy. *Technological Forecasting and Social Change* **90**, 45-61 (2015). [https://doi.org:https://doi.org/10.1016/j.techfore.2013.09.020](https://doi.org/https://doi.org/10.1016/j.techfore.2013.09.020)
- 4 Harmsen, M. *et al.* Integrated assessment model diagnostics: key indicators and model evolution. *Environmental Research Letters* **16**, 054046 (2021). [https://doi.org:10.1088/1748-9326/abf964](https://doi.org/10.1088/1748-9326/abf964)
- 5 Krey, V. *et al.* Looking under the hood: A comparison of techno-economic assumptions across national and global integrated assessment models. *Energy* **172**, 1254-1267 (2019). <https://doi.org:https://doi.org/10.1016/j.energy.2018.12.131>
- 6 Saltelli, A. Making best use of model evaluations to compute sensitivity indices. *Computer Physics Communications* **145**, 280-297 (2002). [https://doi.org:https://doi.org/10.1016/S0010-4655\(02\)00280-1](https://doi.org:https://doi.org/10.1016/S0010-4655(02)00280-1)
- 7 Sobol, I. M. Sensitivity Estimates for Nonlinear Mathematical Models. *Mathematical Modelling and Computational Experiments* **4**, 407-414 (1993).
- 8 Kriegler, E. *et al.* The role of technology for achieving climate policy objectives: overview of the EMF 27 study on global technology and climate policy strategies. *Climatic Change* **123**, 353-367 (2014). [https://doi.org:10.1007/s10584-013-0953-7](https://doi.org/10.1007/s10584-013-0953-7)

Reviewer Reports on the First Revision:

Referees' comments:

Referee #2 (Remarks to the Author):

This manuscript has been significantly revised per the first round of reviews, and in my opinion significantly improved in both (1) clarity and (2) comprehensiveness of the details provided. My initial concerns about the domain specificity and complexity being more appropriate for a more technical journal, as opposed to Nature, still hold to some degree but I am happy to go with consensus of the other reviewers and editor on this point and support publication after some further revision. I will focus comments below on reviews of my original points, roughly in order of appearance in the document.

1. Line 62: Admittedly a picky comment, but it stood out and is in the introductory paragraph so seemed worth mentioning: the term “effective” seems misplaced. Effective at doing what? I would either explain what the authors mean here, or choose a different term.
2. Lines 63-67: The caveat about a large bias towards “a few high abundant models” here feels important to include, as done, and I believe is revisited in the Implications for Policy and Research section (paragraph starting on Line 282) , but I found it slightly confusing here. Are these lines referring to the model differences dimension, or the scenario assumptions dimension? Is this stating a previous weakness that this paper was able to overcome, or foreshadowing a limitation due to data/model/scenario availability as described in the Implications for Policy and Research section (paragraph starting on Line 282) and highlighted by Reviewer 1?
3. Lines 74-79: The manuscript would benefit from a just a few lines summarizing Sobol and variance decomposition here, and perhaps the authors could point to other places Sobol analysis has been applied in the climate change space?
4. Methods (General) and Methods (Variance Decomposition): I am a bit confused here on terminology. Line 462-463 describe decomposition into first and second order variance contributions, and then line 464-465 describes “three indices” that “add up to 1”. If I am not mistaken, first-order Sobol indices do not sum to 1 unless there are no higher order interactions, and total-order similar will sum to more than one if there are higher order interactions, so I am not sure which three would add to 1. Reading the Variance Decomposition portion of the methods, I believe the authors are referring to the three terms computed in 550 – a pointer there and a short summary up front would be helpful to prevent confusion, especially since this computation of indices is a bit non-traditional compared to the classic Sobol decomposition.
5. Other Scenario Assumptions: I appreciate the emphasis on the “second main merit” starting on Line 282, which helps to explain why the authors chose to keep the Other Scenario Assumptions dimension in the main figures, and the nuances of how to interpret the lack of dominance there. This is a subtle part of the paper and the reader benefits from explicit space devoted to it. I still struggle a little bit to understand concretely what this dimension represents, especially in contrast to model

and climate target which can be more explicitly represented as done in SI A Table S1 and S3. Perhaps the authors could provide a few more explicit examples of what this dimension is, even if they cannot be completely enumerated? Another option could be to point to, or summarize in the main text, more of this concept of a noise term that comes up in the Methods and was very helpful to get a better intuition for what this third dimension represents.

Referee #3 (Remarks to the Author):

Authors have sufficiently addressed my concerns. I'm happy to recommend for publication.

Referee #4 (Remarks to the Author):

The authors have addressed all of Referee #1's comments from the technical perspective. However, as other referees have also noted in their comments, the authors should be aware that Nature article is meant for audiences from very diversified background instead of readers who are specialised in IAM modelling.

Here are my specific concerns and comments:

1. The manuscript under review evaluates the 'consensus' and 'robustness' of climate scenario outcomes. However, the interpretation of these terms remains vague throughout the paper. Additionally, it is worth noting that the abstract does not explicitly mention 'consensus' and 'robustness' but instead refers to 'large uncertainties.' Therefore, it is crucial to emphasize the contribution to 'consensus' and 'robustness' in the abstract to provide a clearer representation of the study's findings.
2. The manuscript establishes the concept of 'high consensus' as 'the scenarios results distinctly vary across climate targets and less across model differences and scenario assumptions'. However, it is necessary to clarify how this high consensus is specifically observed in the results of the authors' analyses. The manuscript should provide a detailed explanation of how this consensus is reflected in their findings and interpretations.
3. The manuscript examines three main drivers: climate target, model differences, and other scenario assumptions. In Figure 1(b) and Figure 1(c), the authors appropriately use 'model' as the x-axis to represent 'model differences.' Similarly, in Figure 1(g) and Figure 1(f), 'climate target' is used as the x-axis to represent the variation in climate targets. However, I share your confusion regarding the use of 'climate target' as the x-axis to characterize 'other scenario assumptions' in Figure 1(d) and Figure 1(e). The manuscript should provide a clear explanation or justification for this approach to enhance the understanding of readers. Alternatively, the authors should consider using a more appropriate axis label that accurately represents the variations in other scenario assumptions, ensuring consistency across all figures. Figure 2 and Figure 3 have the same problem.
4. Manuscript Audience : The manuscript demonstrates a high level of technical rigor and is

primarily targeted towards the modelling community specializing in Integrated Assessment Models (IAMs). However, it is crucial for the authors to also consider the broader value of this work for general researchers engaged in climate change modelling and scenario application analysis. Specifically, they should address how this analysis of sensitivity in multi-model assessments enhances general applied work in the field. By highlighting the practical implications and broader applications of their findings, the authors can effectively bridge the gap between the technical aspects of their research and its relevance for a wider audience involved in climate change research and scenario analysis.

Author Rebuttals to First Revision:

Below we present a point-by-point reaction to the comments from the reviewers (in **black**): in **blue** our response and in **orange** the associated changes in the analysis and/or manuscript. For reference, we numbered all reviewer comments, marked in square brackets.

Referee #2 (Remarks to the Author)

This manuscript has been significantly revised per the first round of reviews, and in my opinion significantly improved in both (1) clarity and (2) comprehensiveness of the details provided. My initial concerns about the domain specificity and complexity being more appropriate for a more technical journal, as opposed to Nature, still hold to some degree but I am happy to go with consensus of the other reviewers and editor on this point and support publication after some further revision. I will focus comments below on reviews of my original points, roughly in order of appearance in the document.

We kindly thank the reviewer for his/her comments, which, also in the previous iteration, helped to improve the manuscript significantly. We took the comment on domain specificity and complexity seriously, and addressed this in the following manners (see also #1.2, #1.3, #1.5, #2.1, #2.2, #2.3 and #2.4):

- On domain specificity – We now spell out the general implications much more clear in the conclusions section (in the three ‘merits’, and in the ending paragraph), which clarifies their importance beyond the domain of climate policy modeling.
- On complexity – We have put significant effort in introducing the concepts of consensus, robustness, Sobol’s method and the third (‘other scenario assumptions’) dimension in the introduction, including references previous sustainability papers in *Nature* family journals using these methods and intuitive examples of how a general reader should understand these principles.

[#1.1] Line 62: Admittedly a picky comment, but it stood out and is in the introductory paragraph so seemed worth mentioning: the term “effective” seems misplaced. Effective at doing what? I would either explain what the authors mean here, or choose a different term.

By ‘effective’ climate policy, we were referring to policy instruments (like taxes, regulatory measures or subsidies) that lead to reductions in greenhouse gas emissions. However, we acknowledge that this term is confusing.

The term ‘effective’ is removed, and the sentence has been rewritten to improve clarity.

[#1.2] Lines 63-67: The caveat about a large bias towards “a few high abundant models” here feels important to include, as done, and I believe is revisited in the Implications for Policy and Research section (paragraph starting on Line 282) , but I found it slightly confusing here. Are these lines referring to the model differences dimension, or the scenario assumptions dimension? Is this stating a previous weakness that this paper was able to overcome, or foreshadowing a limitation due to data/model/scenario availability as described in the Implications for Policy and Research section (paragraph starting on Line 282) and highlighted by Reviewer 1?

In the introduction, we mention the large influence of a small set of models as a current problem with how the AR6 scenario database is commonly treated, which is in fact a general problem in climate change mitigation research. 49% of the entries in the AR6 scenario database are produced by only two models. This becomes worse when looking at Paris-aligning scenarios: out of the 212

scenarios that have an end-of-century temperature of 1.5°C, 85 (40%) are from the REMIND(-MAgPIE) model and 63 (30%) are from the MESSAGE(ix-GLOBIOM) model.

In our work, we perform an analysis that *circumvents* this bias, so that we can fairly and more accurately address the importance of model differences as well as the importance of scenario- and climate target differences. In other words, this bias is filtered out and not affecting any of the triangle corners. This is in fact also one of the contributions of this work (indeed a ‘previous weakness that this paper was able to overcome’). The way we do so is by creating uniform samples with an equal number of scenario entries, for each climate target (represented by C1, C2, etc.) and for each model. We determine the relative contributions of the three corners (climate, model and scenario), and redo this process many times – leading to a robust convergence of the results (see SI A.4) that is independent of ‘model-abundance bias’. The process of creating this sample is discussed in the middle of the Methods section.

The reviewer mentions a paragraph in the ‘Implications for Policy and Research’ section in which we describe that we have too little scenario differentiation on a scientific community-wide scale. This is a different problem than the aforementioned model-abundance bias, but associated with, for instance, having a limited range of different socioeconomic pathways or assumptions about availability of technologies. One could also refer to this as a bias – it is like a community-wide or structural bias in terms of scenario thinking – but stemming from a different (although arguably related) problem than the abundance of certain models in the database.

So while the model-abundance bias is something we accounted for, the detection of missing scenario differentiation is more of a *result* of this study. The next step – finding the exact types of scenarios or pathways that climate change mitigation research is missing – is in fact an interesting avenue for a different paper that we hope to pursue in the near-future. In the present paper, we *detected* this problem, which is an important message in itself.

We agree that the distinction between the model-abundance bias and the limited scenario differentiation was not clear in the previous version. We have addressed this in the following way:

- We now explicitly clarify in the ‘Implications for Policy and Research’ section that we are referring to the lacking scenario differentiation rather than model bias (in the ‘second merit’).
- We now include a sentence in the introduction (in the end of the fourth paragraph) to make clear that we account for the model-abundance bias.

[#1.3] Lines 74-79: The manuscript would benefit from a just a few lines summarizing Sobol and variance decomposition here, and perhaps the authors could point to other places Sobol analysis has been applied in the climate change space?

The respective paragraph now includes an accessible and intuitive explanation of what Sobol (and variance) decomposition is, and refers to papers in sustainability literature that also use Sobol’s method^{1,2}.

[#1.4] Methods (General) and Methods (Variance Decomposition): I am a bit confused here on terminology. Line 462-463 describe decomposition into first and second order variance contributions, and then line 464-465 describes “three indices” that “add up to 1”. If I am not mistaken, first-order Sobol indices do not sum to 1 unless there are no higher order interactions, and total-order similar will sum to more than one if there are higher order interactions, so I am not sure which three would add to 1. Reading the Variance Decomposition portion of the methods, I believe

the authors are referring to the three terms computed in 550 – a pointer there and a short summary up front would be helpful to prevent confusion, especially since this computation of indices is a bit non-traditional compared to the classic Sobol decomposition.

The reviewer's interpretation is correct. The three indices defined in this study are a combination of Sobol indices and, as such, are not Sobol indices themselves. This is explicitly written down in the equation that the reviewer is referring to (previously on line 550). While many other papers often focus on first-order Sobol indices, in this particular case, reporting only those may be confusing or misleading. First, because the total effect of a specific driver extends beyond what is merely captured in its first-order effect. Second, when one would present only the *first-order* (Sobol) indices (e.g., in similar triangular figures as we created), readers might mistakenly assume that higher-order effects are negligible. Another argument for including higher-order terms in these indices is the inherent inability to separate the first-order term of the 'other scenario assumptions' dimension. Note that, in the previous review-iteration, we have provided an extensive analysis in SI section A.5, specifically exploring the impact of higher-order terms.

In the sentence pointed out by the reviewer (located at the beginning of the Methods section), we have made it explicit that the indices that 'add up to 1' are a combination of (all) conventional Sobol indices.

[#1.5] Other Scenario Assumptions: I appreciate the emphasis on the "second main merit" starting on Line 282, which helps to explain why the authors chose to keep the Other Scenario Assumptions dimension in the main figures, and the nuances of how to interpret the lack of dominance there. This is a subtle part of the paper and the reader benefits from explicit space devoted to it. I still struggle a little bit to understand concretely what this dimension represents, especially in contrast to model and climate target which can be more explicitly represented as done in SI A Table S1 and S3. Perhaps the authors could provide a few more explicit examples of what this dimension is, even if they cannot be completely enumerated? Another option could be to point to, or summarize in the main text, more of this concept of a noise term that comes up in the Methods and was very helpful to get a better intuition for what this third dimension represents.

We acknowledge that the interpretation of the 'other scenario assumptions' dimension is the most challenging among the three dimensions (model, climate, scenario) to grasp intuitively. In short, this dimension encompasses all factors other than the model used and the climate target set. They range from socio-economic assumptions (e.g., population and GDP) to technological assumptions (e.g., associated with hydrogen, bioenergy and carbon capture and storage, CCS) and even scenario-specific narratives, normative descriptions or mechanisms (e.g., changes in food consumption or trade patterns).

In order to make the 'other scenario assumptions' dimension more clear, we added the SI sections A.5 and A.6 in the previous revision round. Still, however, we agree that these clarifications did not have a large role in the main text yet. For this reason, we now introduce this dimension better in the main text, as well.

Following the reviewer's suggestions, we added the following:

- In the introduction at the end of the second paragraph, we provide a more elaborate explanation of this dimension (using the enumeration above), including the examples given in our response above (in blue).
- We added a motivation for including this dimension in the introduction (i.e., related to, as the reviewer already noticed, the 'second merit' in the conclusions).

- We added references to the Methods and SI sections A.5 and A.6 in the main text.

Referee #3 (Remarks to the Author)

Authors have sufficiently addressed my concerns. I'm happy to recommend for publication.

We thank the reviewer for the previous comments and the recommendation for publication.

Referee #4 (Remarks to the Author):

The authors have addressed all of Referee #1's comments from the technical perspective. However, as other referees have also noted in their comments, the authors should be aware that Nature article is meant for audiences from very diversified background instead of readers who are specialised in IAM modelling.

We kindly thank the reviewer for evaluating reviewer #1's comments and for his/her own comments. The comments helped to improve the manuscript significantly. Specifically, we have reworked the paper to ensure its accessibility to all readers and emphasize the implications beyond the domain of IAM modelling.

These changes can be found in the introduction (before 'Electricity generation') and conclusion sections ('Implications for policy and research'), mainly. See also #1.2, #1.3, #1.5, #2.1, #2.2, #2.3 and #2.4. We introduce a number of technical concepts better, such as 'consensus', the methodology and the 'other scenario assumptions' dimension. In addition, we indicate how the three merits in the conclusion section is relevant for non-modelers such as policymakers and other users of climate change mitigation scenarios.

Here are my specific concerns and comments:

[#2.1] The manuscript under review evaluates the 'consensus' and 'robustness' of climate scenario outcomes. However, the interpretation of these terms remains vague throughout the paper. Additionally, it is worth noting that the abstract does not explicitly mention 'consensus' and 'robustness' but instead refers to 'large uncertainties.' Therefore, it is crucial to emphasize the contribution to 'consensus' and 'robustness' in the abstract to provide a clearer representation of the study's findings.

We agree that the terminology of 'consensus' and 'robustness' were not sharply defined and that a reference to them in the abstract was missing.

Changes:

- In the second and fourth paragraph of the introduction, we introduce the term 'consensus', by indicating that the relative importance of the three drivers (model, climate, scenario) provides insights into the level of agreement (i.e., 'consensus') on the projections of a certain technology or energy carrier. In the paper, we refer to consensus on whether a variable is a robust element of mitigation when the climate targets are driving its variation.
- The abstract now contains a reference to the detection of consensus on elements of climate policy.

[#2.2] The manuscript establishes the concept of 'high consensus' as 'the scenarios results distinctly vary across climate targets and less across model differences and scenario assumptions'. However, it is necessary to clarify how this high consensus is specifically observed in the results of the authors'

analyses. The manuscript should provide a detailed explanation of how this consensus is reflected in their findings and interpretations.

We agree that this was missing in the previous version of the manuscript's main text.

Changes (also in light of comment #2.1):

- We now clarify what is meant by 'consensus' (i.e., a driver of variation being more impactful than model differences). In particular, in this paper, we focus on consensus on whether models agree that a variable is a robust element of mitigation strategy: i.e., the climate target is the dominant (>50%) driver. This is spelled out at the end of the introduction section.
- We added a reference to the Methods, for more information on the interpretation of 'consensus' and the indices in general.

[#2.3] The manuscript examines three main drivers: climate target, model differences, and other scenario assumptions. In Figure 1(b) and Figure 1(c), the authors appropriately use 'model' as the x-axis to represent 'model differences.' Similarly, in Figure 1(g) and Figure 1(f), 'climate target' is used as the x-axis to represent the variation in climate targets. However, I share your confusion regarding the use of 'climate target' as the x-axis to characterize 'other scenario assumptions' in Figure 1(d) and Figure 1(e). The manuscript should provide a clear explanation or justification for this approach to enhance the understanding of readers. Alternatively, the authors should consider using a more appropriate axis label that accurately represents the variations in other scenario assumptions, ensuring consistency across all figures. Figure 2 and Figure 3 have the same problem.

We realize that subpanels Fig. 1(d) and 1(e) are causing some confusion. These subpanels do not have the 'other scenario assumptions' on the x-axis (nor is 'climate target' being used to show this) because we chose to focus on the other two drivers (model and climate target). Also in Fig. 2, we intentionally never use the 'other scenario assumptions' dimension explicitly in the subpanels, nor are we aiming to. The location of the subpanels in Fig. 1 and 2 (including the tables in Fig. 3) is arbitrary and not related to the decomposition triangle (panel (a)). There are a few reasons for this:

- These subpanels are there to show how the dominant driver indeed unravels the variation of the respective variable – e.g., in Fig. 1d, we see that biomass with CCS is mainly varied along climate targets, with the additional insight that this is mainly for targets higher than C3.
- In fact, one of the main messages of the paper is that so little of the scenario spread is driven by other scenario assumptions (the 'second merit' in the conclusions section) – this means that none of these subpanels focus on the 'other scenario assumptions' dimension.
- A label for the x-axis in such a panel is simply missing – we do not have a single label defining how these other scenario assumptions differ (see Methods and SI A.6 on the definition of this dimension). Theoretically we could distinguish the SSP-baselines used in each scenario, but the 'other scenario assumptions' dimension encompasses more than only the socio-economic trajectories.

Note that in Fig. 3, we do show the importance of other scenario assumptions (last column in the tables) in determining various elements of climate policy scenarios, marked as 'Other'.

To avoid the confusion, we now added a note to the caption of Fig. 1 explaining that the 'other scenario assumptions' driver is not focused on in the subpanels.

We also realize that the 'other scenario assumptions' driver can cause confusion. To address this, we provide a more intuitive explanation of this driver, also in line with reviewer comment #1.5.

- In the introduction (end of second paragraph), we provide a more elaborate explanation of this dimension (using the enumeration above), including examples.
- We added a motivation for including this dimension in the introduction (i.e., related to, as the reviewer already noticed, the ‘second merit’ in the conclusions: detecting the sufficiency of current scenario differentiation).
- We added references to the Methods and SI sections A.5 and A.6 in the main text.

[#2.4] Manuscript Audience: The manuscript demonstrates a high level of technical rigor and is primarily targeted towards the modelling community specializing in Integrated Assessment Models (IAMs). However, it is crucial for the authors to also consider the broader value of this work for general researchers engaged in climate change modelling and scenario application analysis. Specifically, they should address how this analysis of sensitivity in multi-model assessments enhances general applied work in the field. By highlighting the practical implications and broader applications of their findings, the authors can effectively bridge the gap between the technical aspects of their research and its relevance for a wider audience involved in climate change research and scenario analysis.

We acknowledge and understand this comment. Therefore, we have restructured part of the introduction, added a number of clarifications to limit the complexity of the paper, and highlighted the practical implications of this work.

Changes:

- The introduction now contains a number of additional clarifications, aimed at making the paper more accessible to a broader (also less-technical) audience:
 - The concepts of ‘consensus’ and ‘robustness’ are now explained more clearly (see also #2.1). We do so both from a qualitative point of view, indicating that ‘consensus’ among model projections is reflected in the level of scenario spread (second paragraph of the introduction), as well as from a quantitative point of view, mentioning how we recognize this in the analysis and figures (final paragraph of the introduction).
 - The methodology (Sobol’s method) is now introduced in an intuitive and accessible manner in the fourth paragraph of the introduction (see also #1.3). Using the impact of model differences on solar power projections as an example, we explain the principles of the methods and the importance of bias. In addition, we refer to existing papers in sustainability science that also use Sobol’s method.
 - The ‘other scenario assumptions’ dimension is now introduced better in the second paragraph of the introduction (see also #1.5), when discussing all three drivers. We provide the examples of such scenario assumptions. In addition, in the fourth paragraph of the introduction, we mention why it is important to address this driver (i.e., linking to the general debate in mitigation literature on scenario differentiation), and provide references to the respective Methods and SI sections.
- The conclusions now more clearly list the most important implications for this work that are relevant for a broad audience:
 - (1) Identifying consensus in climate policy scenarios, which provides a foundation for informed policymaking on many sustainability matters.
 - (2) Scenario differentiation (e.g., also researching post-growth) is crucial to get the grasp of the deep uncertainty associated with climate change.
 - (3) Uncertainty and (model) bias remain significant challenges in state-of-the-art climate science, including that assessed by IPCC. Our research successfully pinpoints

areas where this problem is most pronounced and offers tools to the community to help recognize and tackle this problem.

- The paper's main text now ends with addressing a broader audience rather than solely modelers, as was the case in the previous version. Specifically, this section now echoes and re-summarizes the three general implications discussed above.

References

- 1 Eker, S., Reese, G. & Obersteiner, M. Modelling the drivers of a widespread shift to sustainable diets. *Nature Sustainability* **2**, 725-735 (2019). <https://doi.org:10.1038/s41893-019-0331-1>
- 2 van der Wijst, K.-I., Hof, A. F. & van Vuuren, D. P. On the optimality of 2°C targets and a decomposition of uncertainty. *Nature Communications* **12**, 2575 (2021). <https://doi.org:10.1038/s41467-021-22826-5>

Reviewer Reports on the Second Revision:

Referees' comments:

Referee #2 (Remarks to the Author):

Authors have sufficiently addressed my concerns.

Referee #4 (Only left remarks to the editor):

The referee signs off.

Author Rebuttals to Second Revision:

*Below we present a point-by-point reaction to the comments from the reviewers (in **black**): in **blue** our response and in **orange** the associated changes in the analysis and/or manuscript. For reference, we numbered all reviewer comments, marked in square brackets.*

Referee #2 (Remarks to the Author)

Authors have sufficiently addressed my concerns.

We kindly thank the reviewer for his/her comments, which in previous iterations helped to improve the manuscript significantly.

Referee #4 (Only left remarks to the editor):

The referee signs off.

We kindly thank the reviewer for his/her comments, which in previous iterations helped to improve the manuscript significantly.